# An In-Depth Study on the Metabolite Profile and Biological Properties of *Primula auriculata* Extracts: A Fascinating Sparkle on the Way from Nature to Functional Applications

**DOI:** 10.3390/antiox11071377

**Published:** 2022-07-15

**Authors:** Inci Kurt-Celep, Dimitrina Zheleva-Dimitrova, Reneta Gevrenova, Abdullahi Ibrahim Uba, Gokhan Zengin, Evren Yıldıztugay, Carene Marie Nancy Picot-Allain, José Manuel Lorenzo, Mohamad Fawzi Mahomoodally, Domenico Montesano

**Affiliations:** 1Department of Pharmacognosy, Faculty of Pharmacy, Yeditepe University, 34755 Istanbul, Turkey; incikurt00@gmail.com; 2Department of Pharmacognosy, Faculty of Pharmacy, Medical University-Sofia, 1000 Sofia, Bulgaria; dimizheleva@gmail.com (D.Z.-D.); rgevrenova@gmail.com (R.G.); 3Department of Molecular Biology and Genetics, Faculty of Engineering and Natural Sciences, Kadir Has University, 34083 Istanbul, Turkey; abdullahi.iu2@gmail.com; 4Physiology and Biochemistry Research Laboratory, Department of Biology, Science Faculty, Selcuk University, 42130 Konya, Turkey; 5Department of Biotechnology, Science Faculty, Selcuk University, 42079 Konya, Turkey; eytugay@gmail.com; 6Department of Health Sciences, Faculty of Medicine and Health Sciences, University of Mauritius, 80837 Réduit, Mauritius; picotcarene@gmail.com (C.M.N.P.-A.); f.mahomoodally@uom.ac.mu (M.F.M.); 7Centro Tecnológico de la Carne de Galicia, Rúa Galicia No. 4, Parque Tecnológico de Galicia, San Cibrao das Viñas, 32900 Ourense, Spain; jmlorenzo@ceteca.net; 8Área de Tecnoloxía dos Alimentos, Facultade de Ciencias, Universidade de Vigo, 32004 Ourense, Spain; 9Department of Pharmacy, University of Naples Federico II, Via D. Montesano 49, 80131 Naples, Italy

**Keywords:** *Primula*, antioxidant, phenolics, glucosidase, AGE inhibition, phospho-NF-KB, anti-apoptotic pathway

## Abstract

The biological activity of the aerial part and rhizomes of *Primula auriculata* were assessed for the first time. The biological activities (antioxidant properties, enzyme inhibition, and AGE inhibition) as well as the phenolic and flavonoid contents of the ethyl acetate, ethanol, hydro-ethanol and water extracts of *P. auriculata* aerial parts and rhizomes were determined. Cell viability assays and gelatin zymography were also performed for MMP-2/-9 to determine the molecular mechanisms of action. The gene expression for MMPs was described with RT-PCR. The levels of various proteins, including phospho-Nf-κB, BCL-2, BAX, p-53, and cyclin D1 as well as RAGE were measured using Western blot analysis. The hydro-ethanol extract of the aerial part possessed the highest phenolic (56.81 mg GAE/g) and flavonoid (63.92 mg RE/g) contents. In-depth profiling of the specialized metabolites by ultra-high-performance liquid chromatography—high-resolution mass spectrometry (UHPLC-HRMS) allowed for the identification and annotation of 65 compounds, including phenolic acids and glycosides, flavones, flavonols, chalcones, dihydrochalcones, and saponins. The hydro-ethanol extract of the aerial parts (132.65, 180.87, 172.46, and 108.37 mg TE/g, for the DPPH, ABTS, CUPRAC, and FRAP assays, respectively) and the ethanol extract of the rhizomes (415.06, 638.30, 477.77, and 301.02 mg TE/g, for the DPPH, ABTS, CUPRAC, and FRAP assays, respectively) exhibited the highest free radical scavenging and reducing activities. The ethanol and hydro-ethanol extracts of both the *P. auriculata* aerial part and rhizomes exhibited higher inhibitory activity against acetylcholinesterase, while the hydro-ethanol extracts (1.16 mmol ACAE/g, for both the aerial part and rhizomes extracts) were more active in the inhibition of α-glucosidase. After the treatment of an HT-29 colorectal cancer cell line with the extracts, the apoptosis mechanism was initiated, the integrity of the ECM was remodeled, and cell proliferation was also taken under control. In this way, *Primula* extracts were shown to be potential drug sources in the treatment of colorectal cancer. They were also detected as natural MMP inhibitors. The findings presented in the present study appraise the bioactivity of *P. auriculata*, an understudied species. Additional assessment is required to evaluate the cytotoxicity of *P. auriculata* as well as its activity in ex vivo systems.

## 1. Introduction

Plant extracts are known to possess an array of secondary metabolites that exhibit a number of biological activities with potential biopharmaceutical applications. Secondary metabolites, including alkaloids, tannins, flavonoids, and phenolic compounds, amongst others, occur in various parts of the plant. These metabolites vary considerably and depend on genetic and environmental factors such as the season, temperature, humidity, and soil conditions, among others. Pharmacognosy has unveiled the huge potential of plant extracts, which have been used to manage human ailments for decades. Many of such natural products are being revisited as nutraceuticals and are used as template molecules for the discovery of new drugs to treat and/or manage a panoply of diseases [1].

In the last century, the human population has increased dramatically, and hence, people have a burden to attain safe medicines as well as food. From this point on, the search for new and safe raw materials is essential to continue living healthy. Among the alternative raw materials, unexplored or underutilized plant species are of great interest and are considered promising treasures [2,3,4,5]. Wild plant species can provide alternative sources of health-promoting compounds such as secondary metabolites or vitamins, and they could help balance the diet and control serious health problems such as diabetes or obesity [6,7,8]. In light of this information, our group continues to discover new and effective raw materials from nature. In the present study, *Primula auriculata* was selected as a potential raw material.

*Primula* L., the largest genus of Primulaceae, consists of more than 500 species [9]. The members are widely distributed in the mild climate zone of the northern hemisphere [10]. The members of the genus *Primula* are cultivated for ornamental and medicinal purposes in the mild climate zone [11,12,13]. The *Primula auriculata* used in the present study is a rosette plant measuring between 20 and 50 cm height with oblong-lanceolate leaves and bright-purple or violet flowers that mostly bloom during spring [14]. *P. auriculata* inflorescences produce white powders that were used in traditional medicine to prevent eye diseases such as cataracts, trachoma, blindness, and infection [14]. Previous studies showed that the methanol extract of *P. auriculata* exhibited apoptotic activity on human colon cancer cells (HT-29) by activating caspase [15]. The methanol, hydro-methanol, and water extracts of *P. auriculata* leaves and flowers showed antibacterial activity, especially the methanol extract of *P. auriculata* flowers [14]. The extracts also exhibited radical scavenging properties.

However, there is a dearth of scientific studies revealing the detailed phytochemical profile of different plant parts of *P. auriculata* obtained using solvents of different polarities. The aim of this study was to investigate the antioxidant and enzyme inhibitory activities of the ethyl acetate, ethanol, hydro-ethanol, and water extracts of *P. auriculata* aerial parts and rhizomes as well as to perform cytotoxic and gene expression analyses. The current study is a comprehensive investigation of AGE formation and the protein levels involved in the apoptosis pathway, such as phosphorus-NfB, RAGE, BCL-2, BAX, p-53, and cyclin D1, in HT-29 cell lines. In addition, the levels and gene expression of MMP-2/-9 enzymes were assessed using gelatin zymography and RT-PCR. The obtained results could guide further studies on *P. auriculata* and provide valuable information to cultivate it as a potential raw material for agricultural applications.

## 2. Materials and Methods

### 2.1. Plant Material and Preparation of Extracts

*Primula auriculata* L. samples were collected in the flowering season in June 2020 from a wild population (Hadim, Konya, Egrigöl location, 2100 m). One of the authors (Dr. Evren Yildiztugay) identified the plants, and voucher specimens were deposited at the herbarium of Selcuk University in Konya, Turkey. The aerial parts and rhizomes of the plant samples were dried in the shade at room temperature for about seven days before being ground into powder with a grinder. The samples were stored in a dark environment.

In this study, the extracts were prepared using ethyl acetate (EA), ethanol (EtOH), ethanol/water (EtOH/water, 70%), and water. Maceration was employed as the extraction method to obtain the EA, EtOH, and EtOH/water extracts. The solid–solvent ratio was chosen as 1/20, and the ratio was used in our previous papers [16,17,18]. The plant materials (10 g) were macerated overnight at room temperature with 200 mL of these solvents. The maceration process was performed at 250 rpm in a shaking device (Lab Companion SI-300 Benchtop Shaker, MA, USA). Thereafter, the mixtures were filtered using Whatman No.1 filter papers in a vacuum flask (Büchner flask). Finally, the solvents were evaporated from the mixtures using a rotary evaporator (Heidolph Laborota 4000, Schwabach, Germany) under vacuum at 45 °C.

To obtain water extracts, the plant materials (10 g) were kept with 200 mL of boiled water, and then the extracts were centrifuged at 4000 rpm for 10 min at room temperature to separate the solid materials. Then, the mixture was filtered, similar to the other extracts. Finally, the mixture was lyophilized at −80 °C for 48 h (Syclon-10 N, Ningbo Haishu Sklon Electronic Instrument Inc., Zhejiang, Ningbo, China). All extracts were stored at 4 °C until further analysis was required. The extraction yields are given in Table 1.

### 2.2. Profile of Bioactive Compounds

Folin–Ciocalteu and AlCl_3_ assays were used to determine the total phenolic and flavonoid contents, respectively [19]. For the respective assays, results were expressed as gallic acid equivalents (mg GAEs/g dry extract) and rutin equivalents (mg REs/g dry extract).

### 2.3. Chemicals

The reference standards used for compound identification were obtained as follows: protocatechuic acid (1), 4′-hydroxyflavone (6), apigenin (9), luteolin (11), kaempferol (12), quercetin (13), rhamnetin (16), isorhamnetin (17), apigenin 7-*O*-glucoside (22), luteolin 7-O-glucoside (24), kaempferol 3-*O*-glucoside (26), myricitrin (30), isoquercitrin (31), acaciin (33), kaempferol 3-*O*-rutinoside (35), rutin (37), and pectolinarin (38) were obtained from Extrasynthese (Genay, France); caffeic acid (2) and neochlorogenic acid (3) were supplied by Phytolab (Vestenbergsgreuth, Germany); and chlorogenic acid (4) was purchased from Sigma-Aldrich (St. Louis, MO, USA). Acetonitrile (hypergrade for LC-MS), formic acid (HPLC-grade), and methanol (analytical-grade) were purchased from Merck (Darmstadt, Germany). Phospho- NF-_K_β p65 (Ser536) Monoclonal Antibody (T.849.2) was bought from Thermo Fisher Scientific (Waltham, MA, USA). Rabbit monoclonal anti-RAGE antibody (EPR21171), rabbit monoclonal anti-Bcl-2 antibody (E17), rabbit monoclonal anti-Bax antibody (E63), anti-p53 antibody (PAb 240) (ab26), anti-cyclin D1 antibody (SP4) (ab16663), and an anti-beta-actin antibody (mAbcam 8226) loading control (ab8226) were purchased from Abcam (Cambridge, UK). The Roche-brand cell proliferation reagent WST-1 (Cat No: 5015944001) was bought from Sigma-Aldrich (St. Louis, MO, USA). Gibco Dulbecco’s Modified Eagle Medium (DMEM); Gibco Phosphate-buffered saline PBS (10×), pH 7.4; Gibco Fetal Bovine Serum (FBS), certified, heat-inactivated; and Gibco trypsin-EDTA (0.5%), no phenol red were purchased from Thermo Fisher Scientific. Acrylamide/bis-acrylamide 30% solution 2 (Cat No: A3574), gelatin solution (Cat No: G1393), Amersham™ Protran^®^ Premium Western Blotting membranes, nitrocellulose (Cat No: GE10600004), and Sample Buffer, Laemmli 2× Concentrate were purchased from Sigma Aldrich (St. Louis, MO, USA). QuantiTect Primer Assays (Hs_MMP2_1_SG QuantiTect Primer Assay QT00088396 for MMP-2 and Hs_MMP9_1_SG QuantiTect Primer Assay QT00040040 for MMP-9) and a QuantiTect SYBR Green RT-PCR Kit were bought from QIAGEN (Hilden, Germany).

### 2.4. Ultra-High-Performance Liquid Chromatography Coupled with Hybrid Quadrupole-Orbitrap High-Resolution Mass Spectrometry (UHPLC–HRMS)

The mass spectrometer was operated in the negative and positive ESI modes within the *m*/*z* range from 100 to 1000. The other parameters were as follows: spray voltage, 3.5 kV (+) and 2.5 kV (−); sheath gas flow rate, 38; auxiliary gas flow rate, 12; spare gas flow rate, 0; capillary temperature, 320 °C; probe heater temperature, 320 °C; S-lens RF level, 50; scan mode, full MS (resolution 70,000) and MS/MS (17,500).

The chromatographic separation was performed on a reversed-phase column Kromasil EternityXT C18 (1.8 µm, 2.1 × 100 mm) at 40 °C. The chromatographic analyses were run using 0.1% formic acid in water (A) and 0.1% formic acid in acetonitrile (B) as a mobile phase. The flow rate was 0.3 mL/min. The run time was 33 min. The following gradient elution program was used: 0–1 min, 0–5% B; 1–20 min, 5–30% B; 20–25 min, 30–50% B; 25–30 min, 50–70% B; 30–33 min, 70–95%; 33–34 min, 95–5% B. The equilibration time was 4 min [20]. The data were processed by Xcalibur 4.2 (ThermoScientific) instrument control/data handling software. Metabolite profiling using MZmine 2 software was applied to the UHPLC–HRMS raw files of the studied *P. auriculata* extracts.

### 2.5. Determination of Antioxidant and Enzyme Inhibitory Effects

In this study, a variety of methods were used to evaluate the antioxidant properties of the investigated extracts [21]. These tests included the radical scavenging 1,1-diphenyl-2-picrylhydrazyl (DPPH), 2,2′-azino-bis(3 ethylbenzothiazoline) 6 sulfonic acid (ABTS), cupric-ion-reducing antioxidant capacity (CUPRAC), and ferric-ion-reducing antioxidant power (FRAP) assays, the metal-chelating ability (MCA), and the phosphomolybdenum (PDA) assay. The DPPH, ABTS, CUPRAC, and FRAP test results were expressed as mg Trolox equivalents (TE)/g dry extract, while the MCA and PDA test results were expressed as mg EDTAE/g dry extract and mmol Trolox TE equivalents/g dry extract, respectively. The enzymes acetylcholinesterase (AChE), butyrylcholinesterase (BChE), tyrosinase, amylase, and glucosidase were used to test the extracts. When galanthamine was used as a positive control in the AChE tests, the data were analyzed in terms of galanthamine equivalents (GALAE) per gram of dry extract. Kojic acid extract (KAE), a standard inhibitor of tyrosinase, was utilized to measure the results [21,22]. In the amylase and glucosidase inhibition assays, the acarbose equivalents (ACAE) per gram of dry extract were also calculated.

### 2.6. Molecular Docking

The following crystal structures of the target enzymes were retrieved from the protein data bank (https://www.rcsb.org/ (accessed on 1 June 2022)): AChE (PDB ID: 6O52) [23], BChE (PDB ID: 6EQP) [24], amylase (6TP0) [25], tyrosinase (6JU7) [26], glucosidase (7KBJ) [27], and two cancer targets: MMP-9 (4WZV) [28] and BCL-2 (6QGH) [29]. Missing hydrogen atoms were added at a physiological pH of 7.4, correct bond orders were assigned, and missing atoms were added using Biovia Discovery Studio (DS) (Accelrys Software Inc., San Diego, CA, USA, 2012). Ligand geometries were optimized using the ligand preparation toolkit in Biovia DS.

Each ligand, selected based on % abundance, was docked into the catalytic channel of each of the six enzymes and the binding pocket of BCL-2, and the cocrystal ligand coordinates were used to generate a docking grid file in Autodock 4.2 (https://autodock.scripts.edu (accessed on 1 June 2022)) [30]. The docking grid box and grid size were based on the size of the binding pocket of each enzyme in the respective crystal structure. A ligand conformational search was carried out using a Lamarckian genetic algorithm. The program randomly assigned torsion angles to rotatable bonds. The binding energy of each ligand against each of the enzymes was calculated. A visualization of the protein–ligand interactions was created using Biovia DS Visualizer.

### 2.7. Cell Culture

In this study, the commercially available HT-29 human colon cancer cell line was ordered from ATCC (American Type Culture Collection, Manassas, Virginia, Washington, DC, USA). HT-29 cells were grown in a DMEM medium containing high glucose. The medium was also supplemented with 10% fetal bovine serum (FBS), 1% Penicillin (10 U/mL)/Streptomycin (10 mg/mL), 4.5 g/L of L-glutamine, and 3.7 g/L of sodium pyruvate [31,32]. The HT-29 cells were incubated at 37 °C in humidified 5% CO_2_, which is the optimal growth condition for mammalian cells [33]. When the cells were 90% confluent in the growth dishes, they were passaged by removing them from the culture vessel with 0.05% trypsin/EDTA, and they were stocked in the liquid nitrogen tank for further experiments.

### 2.8. Cell Viability and Cytotoxicity Determination with WST-1

In the experimental principle based on a colorimetric measurement, WST-1 tetrazolium dye produces a water-soluble formazan product on the cell surface due to the NADPH in living cells [34]. A WST-1 assay was used to determine the effect of the extracts obtained from the aerial parts and rhizomes of *P. auriculata* at different concentrations (5, 15, 25, 50, 75, and 100 µg/mL) applied to human colon cancer cell line HT-29 cells on cell viability and cytotoxicity. HT-29 cells were seeded in 96-well plates at a density of 5 × 10^4^ cells per well in 100 µL and incubated in a mammalian cell culture incubator for attachment for 24 h. After 24 h, the HT-29 cell was used as a control cell and was not treated with any extract. In all cell culture experiments presented in L. 202-204, all EtOH and EtOH/water extracts were dissolved in serum-free DMEM (SFM DMEM), which is the growth medium for the cells. In this way, any toxicity or background affect is avoided. On the other hand, the aerial parts and rhizomes of *P. auriculata* at different concentrations (5, 15, 25, 50, 75, and 100 µg/mL) were treated with HT-29 cells for 24, 48, and 72 h, and colorimetric measurements were taken separately at each time point. At the determined time points, the cells, whose incubation period with the extracts was completed, were washed with 1x PBS and then incubated for 1 h with 50 µL of cell growth medium containing a 1:10 WST-1 reagent. After 1 h of incubation, the cells were measured at 570 nm with a Thermo Scientific™ Varioskan plate reader [33,35].

### 2.9. Advanced Glycation End Products Inhibition Assay

The method based on the Maillard reaction [36] was used to determine the AGE inhibition activity of EtOH, and EtOH/water extracts were obtained from the aerial parts and rhizomes of *P. auriculata* at a working concentration of 1 mg/mL. Distilled water was used to prepare the working concentration of *P. auriculata*. The experimental procedure was reported in detail in our previous studies, and different concentrations of quercetin were used as a reference material [33,36,37,38,39].

### 2.10. Western Blot

Western blot analysis consists of three stages: the separation of proteins by polyacrylamide gel electrophoresis (PAGE), the transfer of proteins to the membrane, and making the relevant proteins visible and measurable [40]. Western blot was performed to detect the levels of phospho-NF-_K_β (65 kDa), RAGE (43 kDa), BCL-2 (26 kDa), BAX (21 kDa), p-53 (44 kDa), and cyclin D1 (34 kDa) proteins in HT-29 cell culture lysate. Protein samples were separated by 13% sodium dodecyl sulfate-polyacrylamide gel electrophoresis (SDS-PAGE) in cold running buffer. Then, the target proteins (phospho-NF-_K_β, RAGE, BCL-2, BAX, p-53, and cyclin D1) were transferred to a 0.22 µm pore-sized nitrocellulose membrane. After the transfer, the membranes were incubated in 5% bovine serum albumin (BSA) blocking buffer. After the washing steps, the samples were incubated with HRP-labeled secondary antibodies for two hours at room temperature, and the membrane was transferred to a film cassette after the washing steps with 1× TBS-T solution. An ECL kit based on the chemiluminescence method was used for imaging. The chemiluminescent image was acquired using the Bio-RAD ChemiDoc XRS+ system. β-actin (42 kDa) was used as an equal loading protein to detect the levels of target proteins on the membrane. A quantitative analysis of the densities of the proteins was performed using the “Gel” tab of the Image J program (National Institutes of Health, MD, USA) [41].

### 2.11. Gelatin Zymography Assay

The gelatin zymography method was used for the determination of MMP-2 and MMP-9 enzyme activity, which are members of the matrix metalloproteinase family. This technique is based on the principle that 1% gelatin, which is the substrate of the relevant MMP enzyme, is added to the bottom gel, and the samples are running in the cold running buffer. Where MMPs are active, enzymes consume their substrates, developing white lytic bands on the dark gel. These bands were visualized with the help of the Bio-RAD ChemiDoc XRS+ system on the gel stained with Coomassie brilliant blue reagent [35,41]. The intensity of the white lytic bands in the gel was compared with the band intensity of the untreated HT-29 cells, and Image J was used for the detection of MMP band intensity.

### 2.12. Real-Time Polymerase Chain Reaction (RT-PCR)

RT-PCR was used to detect the MMP-2 and MMP-9 gene expression levels that changed after treating HT-29 cells with nontoxic concentrations of the EtOH and EtOH/water extracts (50 µg/mL) obtained from the aerial parts and rhizomes of *P. auriculata*. In this context, mRNA isolation was performed with the help of peqGold trizol at the end of the 48th hour from untreated HT-29 cells and HT-29 cells treated with the extracts. Then, single-stranded DNA was obtained using the commercially available cDNA kit by Qiagen, and gene expressions were determined by combining them with the primers (QT00088396 for MMP-2 and QT00040040 for MMP-9) of the target MMPs under appropriate conditions [41]. The 18sRNA gene was used as a reference gene to detect changes in the target gene levels in the HT-29 cell line.

### 2.13. Statistical Analysis

All assays were performed in triplicate. In the antioxidant and enzyme inhibitory assays, an ANOVA (Tukey’s test) was used to determine the differences in the tested extracts. In this context, a two-way ANOVA with multiple complications with Sidak was used for the WST-1 and Western blot experiments, while a one-way ANOVA with multiple complications with the Sidak statistical test was used for the AGE inhibition, gelatin zymography, and RT-PCR experiments. *p* ≤ 0.05 was accepted as the smallest significant value and was symbolized with *. Additionally, nonsignificant results were determined with ns at the end of the comparison. All the obtained data were analyzed with the Prism 8 versions of the GraphPad (GraphPad Software, Inc. San Diego, CA, USA) statistical analysis program.

## 3. Results and Discussion

### 3.1. Chemical Composition

Table 1 summarizes the results regarding the total phenol and total flavonoid contents of the different extracts from *P. auriculata* aerial parts and rhizomes. The hydro-ethanol extract of the aerial parts possessed the highest phenolic (56.81 mg GAE/g dry extract) and flavonoid (63.92 mg RE/g dry extract) contents. Regarding the extraction from the aerial parts, it is interesting to highlight that the phenol and flavonoid contents of the ethanol extracts or water extracts were significantly lower as compared to the hydro-ethanolic extracts. These findings might be related to the polarity of the polyphenols present in *P. auriculata* aerial parts. The phosphomolybdenum assay was used to evaluate the total antioxidant capacity (TAC) of the extracts of *P. auriculata* aerial parts and rhizomes. Based on the TAC results, the ethyl acetate extract exhibited the highest activity among the aerial part extracts, while the ethanol extract was the most active among the root extracts. Additional antioxidant assays were conducted in order to obtain a comprehensive understanding of the antioxidant properties of the aerial part and rhizomes extracts of *P. auriculata*.

The identification/tentative annotation of the specialized metabolites in the studied extracts was achieved by UHPLC-HRMS. Nontargeted metabolite profiling and processing of UHPLC-HRMS data by MZ mine 2 software were performed. Based on the MS/MS accurate masses, the fragmentation patterns, and a comparison with the reference standards and literature data, 65 specialized metabolites, including 5 phenolic acids and glycosides, 17 flavones, 21 flavonols, 4 chalcones, 3 dihydrochalcones, 12 saponins, and 3 others, were identified or tentatively annotated.

#### 3.1.1. Phenolic Acids and Glycosides

Compounds 1–4 were unambiguously identified by a comparison of standard references as protocatechuic, caffeic, neochlorogenic, and chlorogenic acids, respectively (Table 2).

Compound 5 [M-H+Formic acid]^−^ at *m*/*z* 521.152, and [M-H]^−^ at *m*/*z* 475.146 gave a base peak at *m*/*z* 293.088 (C_11_H_17_O_9_) and a fragment ion at *m*/*z* 131.033 (C_11_H_17_O_9_-162.05 Da), corresponding to the disaccharide primeverose. Fragment ions at *m*/*z* 181.050 (C_9_H_9_O_4_) and 166.026 (C_9_H_9_O_4_-CH_3_) could be related to methyl 2-hydroxy-4-methoxybenzoic acid. Thus, compound 5 could be ascribed to primeverin/primulaverin, previously isolated from *Primula* species (Table 2) [42]. The total ion chromatogram in negative ion mode, the extracted ion chromatogram, and the MS/MS fragmentation of compound 5 are presented in Appendix A.

#### 3.1.2. Flavones and Flavonols

The flavones 6–11, 14, 15, and 18 and flavonol aglycones 12, 13, 16, and 17 were deduced from the retro-Diels–Alder (RDA) rearrangements ^1,3^A^−^, ^0,4^A^−^, ^1,2^A^−^, ^1,3^B^−^, and ^1,2^B^−^. A variety of *O*-glycosides, including 2′-hydroxyflavone, 7,4′-dihydroxyflavone, apigenin, luteolin, kaempferol, quercetin, myricetin, and acacetin derivatives were identified or tentatively annotated in the studied extracts. Hexosyl, deohyhexosyl, pentosyl, and acetylhexosyl moieties were witnessed by the neutral losses of 162.053, 146.058, 132.042, and 204.064 Da, respectively [43,44]. Compound 32 gave a fragment ion at *m*/*z* 415.103 and a base peak at *m*/*z* 253.050, corresponding to the consecutive losses of ethyl maleic acid and hexose (Appendix A). The fragmentation pathway of the aglycone afforded fragment ions at *m*/*z* 225.053 [Agl-CO]^−^, 209.060 [Agl-CO_2_]^−^, 133.028 (^0,3^B^−^), and 117.033 (^1,3^B^−^) and corresponded to 7,4′-dihydroxyflavone. Thus, 32 was related to 7,4′-dihydroxyflavone *O*-hexoside-*O*-ethylmaleate (Table 2). In addition, seven diglycosides (33–39) and four triglycosides (40–43) were tentatively identified in the studied *Primula* extracts (Table 2). Compounds 6, 9, 11, 12, 13, 16, 17, 22, 24, 26, 30, 31, 33, 35, 37, and 38 were identified by comparison with authentic standards.

#### 3.1.3. Chalcones and Dihydrochalcones

Compound 44 [M-H]^−^ at *m*/*z* 223.075 gave fragment ions at *m*/*z* 205.064 and 195.080, corresponding to the losses of H_2_O and CO, respectively. Fragments at *m*/*z* 119.048 and 117.033 could be related to the cleavage of ^1,7^B^−^ and ^7,8^A^−^, indicating an OH group on ring B [45]. Hence, compound 44 could be ascribed to hydroxychalcone. Compounds 45, 47, and 49 differed from 44 by one, two, and three OH groups and were annotated as di-, tri-, and tetrahydroxychalcone, respectively. In the same manner, 45, 46, 48, and 50 were related to dihydrochalcones (Table 2).

#### 3.1.4. Saponins

In full-scan MS, two isobars, 51 and 52, gave [M-H]^−^ at *m*/*z* 1103.565 (calculated for C_54_H_87_O_23_, 0.524 and 0.306 ppm, respectively). Exemplified by 51, the fragmentation pathway afforded a series of fragment ions in the dd-MS^2^ spectrum, indicative of the sugar chain composition (Appendix A).

The ions at *m*/*z* 957.502 [M-H-146]^−^ and *m*/*z* 923.504 [M-H-146-162]^−^ resulted from the sequential losses of deoxyhexose and hexose moieties. The fragment ions at 923.504 [M-H-(162 + 18)]^−^ and 553.392 [M-H-(2 × 162 + 146 + 2 × 18 + CO_2_]^−^ suggested both a branched oligosaccharide chain with a terminal hexose and *O*-glycosylation of the primary monosaccharide unit. The presence of hexuronic acid was supported by the neutral loss of CO_2_ in the aforementioned fragment. The loss of 646 Da at *m*/*z* 457.371 could be attributed to the tetrasaccharide (2Hex + dHex + HexA). The prominent ion at *m*/*z* 457.371 corresponded to the deprotonated sapogenin (C_30_H_49_O_3_) and was tentatively annotated as protoprimulagenin, commonly found in the *Primula* saponins [42,46,47]. Consistent with the earlier reports, 51 and 52 were related to 3-*O*-[alpha-L-rhamnopyranosyl-(1 → 2)-beta-d-galactopyranosyl-(1 → 3)]-beta-d-glucopyranosyl-(1 → 2)-beta-d-glucuronopyranosyl-protoprimulagenin A (primulasaponin I) and/or its isomers [47,48].

The MS/MS spectra of four isobars (53–56) with [M-H]^−^ at *m*/*z* 1119.560 (calculated for C_54_H_87_O_24_, 0.369, 0.593, 0.914, and 0.700 ppm, respectively) were acquired. The prominent ions at *m*/*z* 973.505 502 [M-H-146]^−^, 957.507 [M-H-162]^−^, 939.497 [M-H-(162+18)]^−^, 811.446 [M-H-(146 + 162)]^−^, and 775.433 [M-H-(146 + 162 + 2 × 18)]^−^ (53) allowed us to deduce a branched sugar chain. Consistent with the oligosaccharide evidenced in the previous saponins, 53–56 afforded a fragment ion at *m*/*z* 569.385 [M-H-(2 × 162 + 146 + 2 × 18 + CO_2_)]^−^ and 473.364 [M-H-(2 × 162 + 146 + 176)]^−^, suggesting a tetrasaccharide (2Hex + dHex + HexA) and a sapogenin at *m*/*z* 473.364 (C_30_H_49_O_4_). The presence of a fragment ion at *m*/*z* 423.328 [sapogenin-H-CH_4_O_2_]^−^ resulted from the concomitant losses of an epoxide moiety and H_2_O. In addition, the ion at *m*/*z* 407.297 [sapogenin-H-CH_4_O_2_ + CH_4_)]^−^ supported by *m*/*z* 391.301 [sapogenin-H-CH_4_O_2_ + CH_2_ + H_2_O)]^−^ are consistent with an epoxide and two tertiary hydroxyl groups, as seen in priverogenin B [48]. Thus, 53–56 were associated with 3-*O*-[alpha-l-rhamnopyranosyl-(1 → 2)-beta-d-galactopyranosyl-(1 → 3)]-beta-d-glucopyranosyl-(1 → 2)-beta-D-glucuronopyranosyl-priverogenin B (Priverosaponin B) and its isomers.

In the same way, two isobars, 57 and 58, with [M-H]^−^ at *m*/*z* 1135.555 (calculated for C_54_H_87_O_25_, 0.756 and 0.871 ppm, respectively) afforded a prominent ion at *m*/*z* 489.3595 [M-H-(2 × 162 + 146 + 176)]^−^, consistent with the sapogenin anagalligenin A [48]. The structure of the aforementioned sapogenin is in agreement with that of priverogenin B, except for the appearance of an hydroxylated epoxyde moiety. Thus, 57 and 58 were ascribed to 3-*O*-[alpha-l-rhamnopyranosyl-(1 → 2)-beta-d-galactopyranosyl-(1 → 3)]-beta-d-glucopyranosyl-(1 → 2)-beta-d-glucuronopyranosyl-anagalligenin A (Primacrosaponin) and its isomer.

The [M-H]^−^ at *m*/*z* 1161.571 (calculated for C_56_H_89_O_25_) of four isobars (59–62) together with the fragment ions at *m*/*z* 1101.544 [M-H-AcOH]^−^ and 611.402 [M-H-(2 × 162 + 146 + 2 × 18 + CO_2_)]^−^ indicated acetylated saponin and a branched tetrasaccharide identical to that in the previous saponins (Appendix A). The precursor ion yielded a sapogenin at *m*/*z* 515.378 accompanied by a fragment at *m*/*z* 473.364 [Sapogenin-H-42]^−^, suggesting the presence of an acetyl group in the sapogenin, as was previously seen in priverogenin B-22 acetate [42,48]. Accordingly, 59–62 were related to 3-*O*-[alpha-l-rhamnopyranosyl-(1 → 2)-beta-d-galactopyranosyl-(1 → 3)]-beta-d-glucopyranosyl-(1 → 2)-beta- d-glucuronopyranosyl-priverogenin B 22 acetate (priverosaponin B 22-acetate) and its isomers.

Compound 52 dominated all studied profiles in both the aerial parts and rhizomes (Appendix A). Compound 53 was found to be the main isomer among 53–56. Primulasaponin I has been found to be a major saponin in *P. auriculata*, *P. megaseifolia*, *P. longipes*, *P. elatior*, *P. elatior* subsp. meyeri, and *P. veris* [42,46,47,48,49]. Włodarczyk et al. [47] suggested that the roots of the non-pharmacopeial primroses *P. conjugens*, *P. chionantha*, and *P. macrophylla* may be alternative sources of primulasaponin I that exceed the level of 10%. Priverosaponin B 22-acetate has been determined exclusively in *P. veris* roots [42]. Isomers 57 and 58 were minor compounds, especially in infusion and ethanol–aqueous root extract (Appendix A). In contrast, 59–62 were evidenced in the ethanol–aqueous aerial parts extracts.

### 3.2. Antioxidant Properties

Until now, no universal antioxidant assay existed to determine the antioxidant capacity of a plant extract. Therefore, antioxidant capacity is evaluated using multiple assays to assess the full antioxidant picture [50]. In this context, the antioxidant properties of *P. auriculata* extracts were evaluated by various assays. Table 3 presents the free radical scavenging (DPPH and ABTS), reducing power (FRAP and CUPRAC), and metal-chelating assessment of *P. auriculata* extracts. The hydro-ethanol and the ethanol extracts of *P. auriculata* aerial part and rhizomes, respectively, exhibited the highest free radical scavenging and reducing activities. As such, the *P. auriculata* aerial part hydro-ethanol (132.65 and 180.87 mg TE/g dry extract, for DPPH and ABTS, respectively) and *P. auriculata* ethanol rhizomes (415.06 and 638.30 mg TE/g dry extract, for DPPH and ABTS, respectively) extracts showed the highest radical scavenging activity. The free radical scavenging abilities of several extracts from some species of the genus *Primula* are described in the literature. Jaberian et al. [14] studied the antioxidant properties of the methanol and methanol/water extract from the flowers and leaves of *P. auriculata*. In their study, the DPPH scavenging abilities were different in the plant parts, and the flower extracts showed stronger abilities than the leaf extracts. Dehghan et al. [51] investigated the biological properties of several plant extracts, including *P. heterochroma* leaf and root extracts. In their study, the best DPPH scavenging abilities for methanol extracts from the tested parts (inhibition percentage were 95.5% and 95.1% for leaves and roots, respectively) were reported. Consistent with our results, the DPPH scavenging results for *P. heterochroma* were correlated with the total phenolic levels in the extracts. In another study, Alinezhad et al. [52] reported the best DPPH scavenging ability for the water extract of *P. heterochroma* with the lowest IC_50_ value. Tarapatskyy et al. [53] studied the chemical characterization and biological properties of different plant part extracts of *P. veris* with different solvents, and the best ABTS and DPPH scavenging abilities were found in the extracts obtained with 70% ethanol. Overall, in agreement with our results, the polar solvent extracts from *Primula* species showed greater radical scavenging abilities than nonpolar solvent extracts. Recent studies have shown that alcohol and hydroalcoholic extracts have a great potential to extract more phenolic compounds from plant samples [54,55,56]. In addition, several authors reported the free radical scavenging abilities of some *Primula* species [57,58,59].

Regarding the reducing potential of *P. auriculata* extracts, *P. auriculata* aerial part hydro-ethanol (172.46 and 108.37 mg TE/g dry extract, for CUPRAC and FRAP, respectively) and *P. auriculata* ethanol rhizomes (477.77 and 301.02 mg TE/g dry extract, for CUPRAC and FRAP, respectively) extracts showed the highest activity. Interestingly, the radical scavenging and reducing potential of the ethanol extract of the rhizomes was significantly higher than the hydro-ethanol extract of the aerial part, and this finding is in line with the high phenolic content of the ethanol extract of *P. auriculata* rhizomes. In this sense, several authors also pointed out that the reducing power values of some *Primula* species were correlated with their total phenolic contents [53,59,60,61,62]. Furthermore, several compounds (caffeic and chlorogenic acids and apigenin, etc.) identified in tested *P. auriculata* extracts have been reported as significant reducing agents in previous studies [63,64,65,66].

Since transition metals are known to participate in Fenton reactions, which generate free radicals, the chelating capacity of aerial part and rhizomes extracts of *P. auriculata* was evaluated. The results presented herein demonstrated that the water extracts possessed strong chelating ability and that the water extract of *P. auriculata* aerial part (25.21 mg EDTAE/g dry extract) was more active than that of *P. auriculata* rhizomes (21.95 mg EDTAE/g dry extract). The obtained results are in accordance with the results reported by Alinezhad et al. [52] who found that the best metal-chelating ability was provided by aqueous extract of *P. heterochroma*.

### 3.3. Enzyme Inhibitory Properties

Alzheimer’s disease (AD) is a neurodegenerative disease that is characterized by irreversible cognitive decline and neurodegenerative dysfunction due to synapse loss and the accumulation of senile plaques [67]. The pathophysiology of AD has been related to the reduction in acetylcholine neurotransmitter, which is hydrolyzed by acetylcholinesterase. In fact, the deterioration of cholinergic neurons in the brain and the loss of neurotransmission are the major causes of cognitive decline [68]. Limiting acetylcholine hydrolysis by inhibiting acetylcholinesterase has been found to stimulate neurotransmission. As such, agents responsible for the inhibition of cholinesterases, namely, donepezil, rivastigmine, and galantamine, are currently used for the management of AD. Besides, butyrylcholinesterase activity has been reported to progressively increase during the latter stage of AD, while acetylcholinesterase declines or remains unchanged [69]. In this study, the ethanol and hydro-ethanol extracts of both *P. auriculata* aerial part and rhizomes exhibited higher inhibitory activity against acetylcholinesterase. On the other hand, among the *P. auriculata* aerial part extracts, the ethyl acetate extract (3.54 mg GALAE/g dry extract) exhibited the highest activity against butyrylcholinesterase, while the rhizomes extracts (3.39–3.51 mg GALAE/g dry extract) showed considerable inhibition potential, except the water extract (Table 4).

Diabetes mellitus type 2 (DM) is a common metabolic disorder that is strongly related to AD. The relationship between DM and AD was first observed in 1999 in a study that revealed that diabetic patients have a 2-fold higher risk of developing AD [70]. Inhibitors of enzymes, namely, α-amylase and α-glucosidase, which are responsible for the hydrolysis of ingested carbohydrates, are at the frontline of DM management. In this work, the extracts of P. auriculata aerial part and rhizomes were examined for α-glucosidase and α-amylase inhibition, and the results are presented in Table 4. Regarding α-amylase inhibition, the ethyl acetate extracts (0.67 and 0.54 mmol ACAE/g dry extract, for aerial part and rhizomes extracts, respectively) showed a higher inhibition. For α-glucosidase inhibition, hydro-ethanol extracts (1.16 mmol ACAE/g dry extract, for both aerial part and rhizomes extracts) were more active. In fact, the inhibition of α-glucosidase has been claimed to be an ideal strategy for the management of DM since it was reported to exhibit fewer side effects.

The ability of *P. auriculata* to inhibit tyrosinase was also assessed and is presented in Table 4. Tyrosinase is a copper-containing enzyme that is responsible for the biosynthesis of melanin, a brown pigment required for photoprotection. However, overproduction of melanin is associated with skin hyperpigmentation problems, such as lentigines, melasma, freckles, and acanthosis nigricans. Therefore, the inhibition of tyrosinase is regarded as an effective therapeutic strategy for the management of skin hyperpigmentation problems. In the present study, the ethanol and hydro-ethanol extracts of *P. auriculata* aerial part and rhizomes showed the highest inhibition against tyrosinase, while the water extracts exhibited the lowest activity.

### 3.4. Molecular Docking

The calculated binding energy values of each ligand against each of the proteins was tabulated (Table 5). Kaempferol 3-*O*-rutinoside and rutin (quercetin 3-*O*-glucoside) showed good binding potential to all seven proteins, with the tightest binding to AChE and BChE. Myricitrin (myricetin 3-*O*-rhamnoside) and dihydroresveratrol 3-*O*-hexoside bound more favorably to AChE and glucosidase, whereas protocatechuic acid, 4′-hydroxyflavone, hydroxychalcone, and azelaic acid demonstrated modest binding to all the proteins.

Therefore, the detailed protein–ligand interaction was visualized. The major contributors to the interaction between kaempferol 3-*O*-rutinoside and the five enzymes were H-bond interactions with polar amino acid residues via multiple hydroxyl groups. Kaempferol 3-*O*-rutinoside was buried in the catalytic channels of AChE, BChE by forming for interactions compared with glucosidase and amylase, and the least effective was tyrosinase. In addition to H-bond formation, several van der Waals interactions and a couple of hydrophobic interactions contributed to strength of the binding, especially to AChE and BChE (Figure 1A–D).

To investigate the anticancer potential, these compounds were also docked to MMP-9 protein, which play roles in invasion, metastasis, and angiogenesis [71], and BCL-2, which plays roles in cell death by regulating apoptosis [72]. Kaempferol 3-*O*-rutinoside binds well to both MMP-9 and BCL-2 protein by forming multiple H-bonds and several van der Waals interactions (Figure 1E,F).

Similarly, rutin (quercetin 3-*O*-glucoside) exhibited the strongest interaction with both AChE (Figure 2A) and BChE (Figure 2B) by the formation of multiple H-bonds via hydroxyl groups. On the other hand, azelaic acid, being a linear molecule, spanned the catalytic channel of tyrosine by forming two H-bonds and several van der Waals interactions all over the narrow tunnel (Figure 2C). For glucosidase, myricitrin (myricetin 3-*O*-rhamnoside) filled up its active site via multiple π-π interactions in addition to the major H-bonds and several van der Waals interactions (Figure 2D). In a similar manner, primulasaponin displayed strong binding to both MMP-9 and BCL2, possibly contributing to the anticancer activity observed with the extracts (Figure 2E,F).

Taken together, the major compounds in the extracts of *P. auriculata* are likely exerting biological activity by binding to the catalytic channels of AChE, BChE, tyrosinase, amylase, and glucosidase as well as the cancer target proteins MMP-9 and BCL-2.

### 3.5. ADMET Prediction

A prediction of ADMET properties for the most abundant compounds in *P. auriculata* extracts was performed on the SwissADME web server (http://www.swissadme.ch/ (accessed on 1 June 2022)) developed by the Swiss Institute of Bioinformatics to enable the computational estimation of physiochemical descriptors and pharmacokinetic properties. The SMILE code for each compound was uploaded to the webserver, and their ADMET properties were computed. The ADMET plot of logP (logarithm of octanol–water partition coefficient) versus PSA (polar surface area) was generated using Biovia DS.

The ADMET properties of each compound are presented in Table 6. Protocatechuic acid, 4′-hydroxyflavone, hydroxychalcone, and azelaic acid were found to have high gastrointestinal (GI) absorption and could cross the blood–brain barrier. On the other hand, kaempferol 3-*O*-rutinoside, rutin (quercetin 3-*O*-glucoside), myricitrin (myricetin 3-*O*-rhamnoside), primulasaponin, and dihydroresveratrol 3-*O*-hexoside were predicted to have low GI absorption, could not cross the blood–brain barrier due to the high polarity, and have violated some of the Lipinski’s “rule of 5” parameters due to their large size (Figure 3). Nonetheless, all the compounds were predicted to be nontoxic for the liver, as they could not inhibit the liver metabolic CYP450 enzymes.

### 3.6. Molecular Approach in Apoptotic Pathway

As shown in Table 2, Table 3 and Table 4, it has been observed that different extracts obtained from the aerial parts and rhizomes of *P. auriculata*, especially EtOH and EtOH/water samples have rich chemical contents and high antioxidant potentials and enzyme activity. As a result of these experiments, EtOH and EtOH/water extracts obtained from the aerial parts and rhizomes of *P. auriculata* were used in in vitro cell culture experiments to be applied to the HT-29 colon cancer cell line. Although there are preliminary data on the potential of *P. auriculata* against those cell lines, the current study provides detailed information for understanding the molecular mechanisms in cancer progress, such as the proteins, enzymes, and genes participating in apoptotic pathways. Moreover, the AGE/RAGE interaction after the treatment with extracts was also indicated.

#### 3.6.1. Cell Viability and Cytotoxicity Determination with WST-1

Firstly, the WST-1 test, depicted in Figure 4, was performed to determine the nontoxic concentration of EtOH and EtOH/water extracts obtained from the aerial parts and rhizomes of *P. auriculata* in the HT-29 cell line and to determine the concentration to be used in future experiments.

After seeding HT-29 cells at a definite density as described in Section 2.8, EtOH (AP) (Figure 4A), EtOH/water (AP) (Figure 4B), EtOH (rhizomes) (Figure 4C), and EtOH/water (rhizomes) (Figure 4D) extracts were applied to HT-29 cells at six different concentrations for 24, 48, and 72 h. Untreated HT-29 cells were used as the control group, and cell viability was calculated as 100% at each time point. All experiments were repeated three times independently from each other. As presented in Figure 4A, 5 µg/mL and 15 µg/mL EtOH (AP) applied to HT-29 cells after 24, 48, and 72 h did not affect cell viability, and proliferation was duplicated (*p* ≤ 0.0001 ****). On the other hand, it was observed that cell viability decreased in the treatment at 25, 50, 75, and 100 µg/mL concentrations. As a result of the application at a concentration of 75 µg/mL, the cells had a viability of 76.49% for 24 h, 63.49% for 48 h, and 58.22% at the end of 72 h (*p* ≤ 0.0001 ****). When similar comparisons were performed for 100 µg/mL, it was observed that the cells showed 64.91%, 54.23%, and 45.33% viability at the determined time points, respectively (in Figure 4A, *p* ≤ 0.0001 ****). Similar results were obtained as a result of the applications of the EtOH/water (AP) extract obtained from the aerial parts of *P. auriculata* at the same time points (in Figure 4B). It was observed that 5 µg/mL and 15 µg/mL EtOH/water (AP) applied to HT-29 cells did not affect the proliferation time, and the viability increased by approximately 50% when compared to untreated HT-29 cells (*p* ≤ 0.0001 ****). In the same experimental setup, cell viability was shown to decrease due to applications at 25, 50, 75, and 100 µg/mL concentrations (in Figure 4B). At the end of 24, 48, and 72 h of EtOH/water (AP) application at 25 µg/mL, the viability was calculated as 103.26% (*p* > 0.05 ns), 98.12% (*p* > 0.05 ns), and 89.16% (*p* ≤ 0.001 ***), respectively (in Figure 4B). As a result of 50 µg/mL application, cell viability was found to be 90.18% (*p* ≤ 0.01 **), 89.76% (*p* ≤ 0.01 **), and 74.55% (*p* ≤ 0.001 ***), respectively. After the application at 75 µg/mL, these values decreased to 82.64%, 71.12%, and 60.16%, respectively (*p* ≤ 0.0001 ****). In addition, it was found that when performed for 100 µg/mL, the cells showed 70.24%, 68.25%, and 52.37% viability at the determined time points, respectively (*p* ≤ 0.0001 ****). The results are depicted in Figure 4C,D, and the effect of the EtOH and EtOH/water extracts obtained from the rhizomes of *P. auriculata* on HT-29 cell viability showed similar results.

The cell viability was demonstrated to decrease at increasing concentrations in the rhizomes of *P. auriculata*, where the results were similar to the aerial part. These results indicate that both the aerial parts and rhizomes of *P. auriculata* have a cytotoxic effect on HT-29 cells at increasing concentrations and time points, especially at 75–100 µg/mL. According to literature survey, the chemical compositions of these extracts consist of 12 different saponins [73]. Saponins are well-known natural glycosides found in plants and demonstrate cytotoxicity in many cancers, mainly colorectal cancer [73,74,75,76]. They are widely referred to contribute to treatment as a new generation of anticancer agents since they prevent cell growth by prompting cytotoxicity at increasing concentrations, inducing apoptosis and activating the autophagy mechanism [77]. In light of this information, 50 µg/mL was reported as the promising dose in various anticancer studies since it was shown to prevent cell growth without causing less than 50% toxicity in an HT-29 colorectal cancer cell line. In another study, 40 µg/mL methanolic extract prepared from the aerial parts of *P. auriculata* was detected as the nontoxic dose, and it was the leading idea for our study [15]. The nontoxic concentration of the EtOH and EtOH/water extracts obtained from different parts of *P. auriculata* in cancer molecular pathways of the HT-29 cell line was revealed for the first time in this study.

#### 3.6.2. AGE Inhibition Activity of *Primula auriculata*

AGE, a product of the Maillard reaction, is also one of the cancer markers that emerges as a result of the increased amount of cellular ROS [36,78,79]. In this context, it is hypothesized that if the model plant *P. auriculata* of the study has an AGE inhibition effect, it will play a crucial role in the cancerization process and in the prevention of the metastasis of the disease. In Figure 5, the AGE inhibition values of different extracts are exhibited. This experimental setup was repeated at least three times at different times, and quercetin was used as a reference substance. In the results in Figure 5A, it was determined that the highest inhibition activity was 95.13% at 1000 µg/mL, and the lowest inhibition was 40.31% in quercetin at a concentration of 62.5 µg/mL. On the other hand, in EtOH (AP), EtOH/water (AP), EtOH (rhizomes), and EtOH/water (rhizomes) extracts at 1 mg/mL concentration, 60.27%, 72.65% (*p* ≤ 0.001 ***), 92.38% (*p* ≤ 0.0001 ****), and 83.73% (*p* ≤ 0.0001 ****) inhibition rates were observed, respectively (Figure 5B). The EtOH (rhizomes) extract with the highest AGE inhibition was observed to be in parallel with the content analysis and enzyme inhibition experiments conducted within the scope of the study. Even in the EtOH (AP) extract, which induced the lowest inhibition value of 60.27%, it is an acceptable result if the value is more than 50%. Briefly, the AGE-inhibitory potentials of extracts were found to be; EtOH (rhizomes) > EtOH/water (rhizomes) > EtOH/water (AP) > EtOH (AP). Plants with AGE inhibition activity have the potential to be a target natural drug for cancer treatment nowadays [78,80,81,82]. Plants are referred to as natural AGE inhibitors due to their polyphenolic compounds, terpenoids, various vitamins, alkaloids, and various compounds [82]. As shown in Table 2, extracts obtained from both the aerial parts and rhizomes of *P. auriculata* have high antioxidant activity, and their chemical compositions were found to contain various bioactive molecules. These results confirmed our AGE inhibition experiments. Through the AGE inhibition mechanism of *P. auriculata*, increased ROS clearance in cancer cells, the reduction in active carbonyl groups, the chelation of metal ions, and the precipitation of proteins, which are the building blocks of the cell, can be prevented, and the progression of the disease can be stopped [15,82]. The AGE inhibition potential of extracts obtained from different parts of *P. auriculata* was exhibited for the first time with this study, and it was effective in elucidating the molecular mechanism of cancer and in inspiring drug discovery. On the other hand, the current study revealed that *P auriculata*, which was observed to be a natural AGE inhibitor, might affect the synthesis of other proteins contributing to ROS-mediated cancer pathways, and this led to preliminary data for Western blot experiments to elucidate this molecular mechanism.

#### 3.6.3. Western Blot and Protein Analysis

It is well-known that various molecular pathways in cancer progression in HT-29 cells are activated/suppressed by proteins and various cellular enzymes [83,84,85]. The promising results observed in terms of AGE inhibition and the high antioxidant capacities of *P. auriculata* extracts led us to expose the roles of these extracts in ROS-related and AGE-induced protein pathways in HT-29 cells. In this context, nontoxic concentrations of the extracts (50 µg/mL) were applied to colorectal cancer cell lines for 48 h to elucidate the molecular mechanism of the ROS-induced cancer molecular pathway in HT-29 cells. At the end of 48 h, protein isolation was performed from these cells, and then phospho-NF-_K_β, RAGE, BCL-2, BAX, p-53, and cyclin D1 protein levels were monitored by the Western blot method. β-actin was used as a loading control for the equal amount of protein (Figure 6A). The band densities of the protein samples in each well were compared to that of β-actin, and a non-treated HT-29 cell line was selected as the control. Cisplatin (10 µg/mL) was used as a positive control [86]. Statistical evaluations were conducted by assuming the band density of the HT-29 cell in the control group to be 1-fold (Figure 6B).

As depicted in Figure 6B, phospho-NF-_K_β protein localized to the nucleus was detected as 0.30-fold in cells treated with EtOH (AP) and EtOH/water (AP) and 0.56-fold in cells treated with cisplatin (*p* ≤ 0.05 for *). Moreover, the phospho-NF-_K_β protein level was shown to decrease by 70% in extracts obtained from aerial parts compared to HT-29 (*p* ≤ 0.05 for *). Similarly, in HT-29 cells treated with EtOH (rhizomes) and EtOH/water (rhizomes), the phospho-NF-_K_β protein levels were decreased, at 85% (*p* ≤ 0.01 for **) and 81% (*p* ≤ 0.01 for **), respectively. An analogous experimental setup was used to detect the level change in RAGE protein synthesis, and decreases of 59% (*p* ≤ 0.05 for *) for samples with cisplatin, 70% for EtOH (AP), and 74% for EtOH/water (AP) were observed (*p* ≤ 0.05 for *). After that, HT-29 cells were treated with EtOH (rhizomes) and EtOH/water (rhizomes), and RAGE levels were found to drop by 84% and 78%, respectively (*p* ≤ 0.01 for **). When HT-29 cells were treated with 10 µg/mL cisplatin, EtOH (AP), EtOH/water (AP), EtOH (rhizomes), and EtOH/water (rhizomes), the reductions in the levels of BCL-2, a member of the anti-apoptotic protein family, were detected as 33% (*p* ≤ 0.05 for *), 61% (*p* ≤ 0.05 for *), 62% (*p* ≤ 0.05 for *), 83% (*p* ≤ 0.01 for **), and 73% (*p* ≤ 0.01 for **), respectively. On the other hand, the level of apoptosis-inducing BAX protein in cisplatin-treated samples was found to increase 5.81 times (*p* ≤ 0.0001 for ****) compared to the untreated HT-29 cells. Therefore, the anti-apoptotic pathway was suppressed in cells treated with EtOH (AP) and EtOH/water (AP), and these increases were 11.20- and 17.63-fold, respectively (*p* ≤ 0.0001 for ****). After EtOH (rhizomes) and EtOH/water (rhizomes) treatment, the levels BAX protein were detected to increase 22.73 and 10.48 times compared to HT-29 cells. Another protein that plays an active role in the control of the cell cycle is p-53. After 10 µg/mL cisplatin, EtOH (AP), EtOH/water (AP), and EtOH (rhizomes) treatment, the levels of p-53 increased by 4.54 times, 12.48 times, 4.95 times, 14.94 times, and 4.14 times, respectively, compared to the untreated group (Figure 6B). Similarly, cyclin D1 was observed to decrease by 31% (*p* ≤ 0.05 for *) in HT-29 cells treated with cisplatin, 57% (*p* ≤ 0.05 for *) for EtOH (AP)-treated HT-29 cells, and 50% (*p* ≤ 0.05 for *) for EtOH/water (AP)-treated HT-29 cells. The cyclin D1 level was shown to decrease by 83% after the application of EtOH (rhizomes), while there was also a decrease by 53% after EtOH/water (rhizomes) treatment (*p* ≤ 0.01 for **). In the light of these findings, it has been observed that EtOH (AP), EtOH/water (AP), EtOH (rhizomes), and EtOH/water (rhizomes) extracts applied to the human colorectal adenocarcinoma cell line HT-29 inhibit protein pathways that play an active role in the progression of cancerization and encourage cells to controlled division by increasing the synthesis of apoptotic proteins. In normal cells, NF-_K_β is bound to IKB protein in the cytosol and is in an inactive form. Due to factors such as increased ROS, UV exposure, inflammation, and decreased antioxidant levels in the cell, IKB bound to NF-_K_β is phosphorylated by IKK, and NF-_K_β is stimulated to become active in the nucleus [87,88,89]. At the end of the translocation from the cytosol to the nucleus, the increased NF-_K_β protein in the nucleus impairs the hemostasis of the cell, increases cell proliferation, and supports the cells in escaping from apoptosis [88,89,90]. This activated molecular pathway initiates the cancer process of cells. The findings in the Western blot experiment showed that the active form of phospho-NF-_K_β was at the highest level in the untreated HT-29 cell lines, and this level was reduced again after the HT-29 cells were treated with *P. auriculata* extracts. It is understood that the lost cellular ROS/antioxidant balance can be regained due to the highly antioxidant compounds found in *P. auriculata* and that NF-_K_β is reduced in the nucleus and translocated to the cytosol, thus shifting into an inactive form [15,61]. The findings given in Figure 6 support this hypothesis and are in agreement with those in the literature. Another key player in the ROS-mediated cancer process is RAGE, the receptor of AGE. RAGE decreases with embryonic development, and in adulthood, it only remains in the lung cells in trace amounts. It has a molecular weight of approximately 45–50 kDa. It is overexpressed again in processes such as increased ROS, decreased antioxidant level, the suppression of the immune system, and tumor and cancer formation in cells [78,91]. RAGE must bind with AGE, high-mobility group box proteins (HMGB1), S100 calgranulins, and amphoterine in cells for either an immune response or cancer progression. Cell survival and cancer progression have been observed as a result of the binding of RAGE with AGEs. It has been determined that the levels of proteins that activate many signaling pathways such as MAPK, NF-KB, p-53, BCL-2, BAX, and cyclin D1 in cancer cells are changed by the AGE/RAGE relationship [78,91,92]. Various studies have shown that NF-kB, which causes the secretion of proinflammatory cytokines, growth factors, and adhesion molecules, is activated by the AGE-RAGE signal in cancer cells induced by oxidative stress [91,92,93]. In numerous cancer studies, especially those in colorectal colon cancer, AGE/RAGE-driven increased NFKB levels cause cell survival, escape from apoptosis, angiogenesis, and the disruption of the extracellular matrix (ECM). Cancer cells were observed to raise the levels of NFKB-associated anti-apoptotic BCL-2 to escape from apoptosis, while decreasing the levels of p-53 and pro-apoptotic BAX proteins [92]. On the other hand, accumulated data have indicated that the expression of the cell cycle control protein cyclin D1 is increased by NFKB translocated to the nucleus. This regulation triggers the survival mechanism of the cell, meaning that the cell is insensitive to death [94,95,96]. In line with this information, it was observed that the extracts obtained from different parts of *P. auriculata* applied to HT-29 cells inhibited AGE thanks to high levels of antioxidant compounds, and thus, its receptor, RAGE, was inhibited. The NF-_K_β signaling pathway was inhibited by RAGE, which was suppressed by the inhibition of AGE, and the expression of BCL-2 was decreased. By increasing the expression of tumor suppressor p-53 downstream of NF-_K_β, the synthesis of pro-apoptotic BAX protein was increased, and thus, the apoptosis mechanism of cells was initiated. In addition, controlled cell division was promoted in HT-29 cells treated with the extracts by decreasing the level of cyclin D1 responsible for the G1/S cell checkpoint. The Western blot results showed that *P. auriculata* extracts are promising for future use in the treatment of colorectal cancer and can be natural anti-apoptotic inhibitors due to the decreased NFKB, RAGE, BCL-2, and cyclin D1 results.

#### 3.6.4. Matrix Metalloproteinase Activation and Expression

MMPs are known to be overexpressed in cancer cells for metastasis and thus accelerate the cancer spread and invasion process by breaking the ECM composition [97,98,99,100,101]. In the gelatin zymography and RT-PCR assays where similar results were obtained, the roles of *Primula* extracts (50 µg/mL) applied to H-29 cells on MMP-2/-9 were examined. In this context, an experimental setup was installed in which HT-29 cells represented the control group and HT-29 cells treated with 10 μg/mL cisplatin were a positive control. Gelatin zymography (for MMP-2 and -9) and RT-PCR experiments were performed with medium treated for 48 h and DNA samples, respectively. In the experiments for all gelatin zymography, the band density of the untreated HT-29 cell lines was assumed to be 100%, and the statistical analysis was performed accordingly. MMP-9 activity was found to be 45.20% in HT-29 cells treated with 10 µg/mL cisplatin (in Figure 7). The MMP-9 values in cells treated with EtOH (AP), EtOH/water (AP), EtOH (rhizomes), and EtOH/water (rhizomes) were determined to be 38.78%, 44.77%, 28.50%, and 34.11%, respectively (in Figure 7, *p* ≤ 0.0001 ****). When a similar comparison was made for MMP-2 enzyme activity (Figure 8), values of 58.78% for cisplatin, 40.72% for EtOH (AP), 34.89% for EtOH/water (AP), 16.83% EtOH (rhizomes), and 20.62% for EtOH/water (rhizomes) were observed (*p* ≤ 0.0001 ****). In Figure 9, the mRNA levels of MMP-2 and MMP-9 are demonstrated. In Figure 9A, MMP-9 was found to be reduced by approximately 3-fold in HT-29 cells treated with cisplatin, EtOH (AP), and EtOH/water (AP) compared to untreated cells. MMP-9 gene expression was decreased by 4- and 3.23-fold in HT-29 cells treated with EtOH (rhizomes) and EtOH/water (rhizomes), respectively (*p* ≤ 0.0001 ****). The MMP-2 gene expression results given in Figure 9B pointed out that the mRNA level in cisplatin-treated cells was similar to that of EtOH (AP) and EtOH/water (AP)-treated HT-29 and was approximately 2.1-fold less expressed (*p* ≤ 0.0001 ****). The mRNA levels for EtOH and EtOH/water were 2.95-fold and 2.38-fold less expressed, respectively (*p* ≤ 0.0001 ****). Previous studies have shown that increased AGE/RAGE interaction results in a rise in MMP levels, leading to cancer cell metastasis, invasion, and angiogenesis [78,92,93]. Deng et al. found that the level of overexpressed MMP-2 in colorectal cancer cells was reduced after the inhibition of RAGE with a specific siRNA. Thus, metastasis and invasion were prevented, and ECM integrity was regained [102]. On the other hand, Zhu et al. have shown that RAGE-associated HMGB1 is responsible for the upregulation of MMPs, and MMP activation causes cancer cell proliferation via the RAGE/Snail/NF-κB signaling pathways, especially in colorectal cancer [103]. The current study revealed that cancer progression, metastasis, cell proliferation, and invasion can be prevented by inhibiting the ROS-driven AGE/RAGE signaling pathway in the HT-29 colorectal cancer cell line with *P. auriculata* extracts. This study seems to be the first to indicate that *P. auriculata* extracts are natural MMP inhibitors and promising candidates in the search for new anticancer drugs, especially in the prevention of colorectal cancer metastasis.

## 4. Conclusions

The present work provided the first comprehensive insights into the chemical composition and biological properties of *P. auriculata* extracts. The chemical composition and the biological activities clearly depended on the plant parts (aerial parts and rhizomes) and solvents used in the extraction. The data collected from the antioxidant assays showed that ethanol/water and water (infusion) extracts from each part had the strongest abilities. In terms of the enzyme-inhibiting properties, with minor exceptions, the best effects were achieved with the ethanol and hydro-ethanol extracts. In the chemical profiles of the extracts, 65 compounds were identified, more than half of which were flavonoids (38 compounds) and saponins (12 compounds). The high AGE inhibition potential of the ethanol and hydro-ethanol extracts obtained from various parts of *P. auriculata* suppressed the RAGE receptor. In applications with ethanol and ethanol/water extracts from both parts of the plant, nuclear NF-κB, BCL-2, and cyclin D1 levels were reduced, while the levels of BAX and p53 were increased. This finding indicated that the extracts increased the cells’ susceptibility to apoptosis. The levels of overexpressed MMPs responsible for the loss of ECM integrity in HT-29 cells were reduced after *P. auriculata* treatment, and the ECM was remodeled. The results provide new scientific insights into the chemical profiles and biological effects of *P. auriculata*, which may serve as a new and potent source of health-promoting compounds, such as antioxidants and anticancer agents, and the plant could be a potential raw material for the development of functional applications in the nutraceutical and pharmaceutical industries. However, further studies on bioavailability and in vivo testing (particularly animal assays) of the extracts are strongly recommended.

## Figures and Tables

**Figure 1 antioxidants-11-01377-f001:**
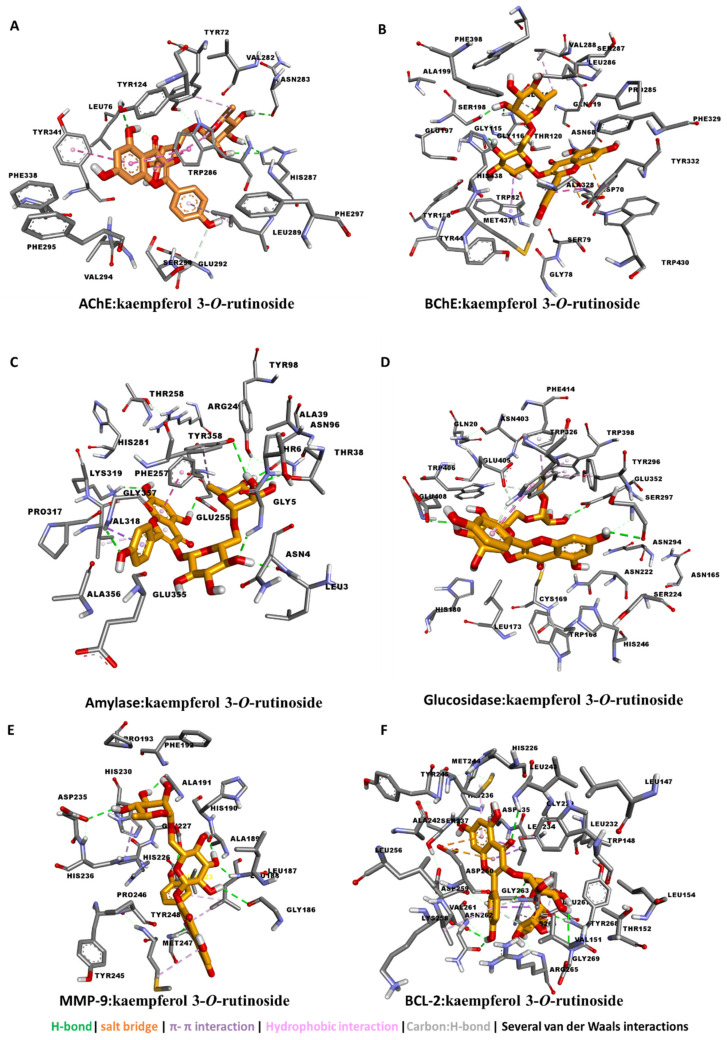
Protein–ligand interaction: kaempferol 3-*O*-rutinoside with (**A**) AChE, (**B**) BChE, (**C**) amylase, (**D**) glucosidase, (**E**) MMP-9, and (**F**) BCL-2.

**Figure 2 antioxidants-11-01377-f002:**
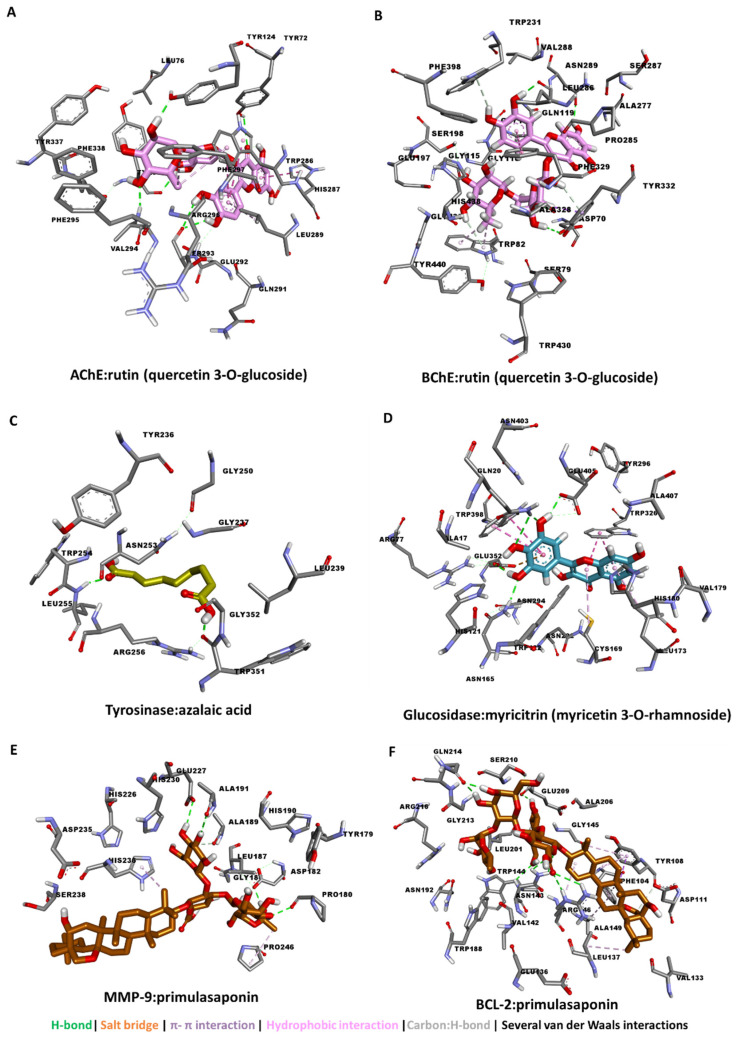
Protein–ligand interaction: (**A**) AChE and rutin (quercetin 3-*O*-glucoside), (**B**) BChE and rutin (quercetin 3-*O*-glucoside), (**C**) amylase and azalaic acid, (**D**) glucosidase and myricitrin (myricetin 3-*O*-rhamnoside), (**E**) MMP-9 and primulasaponin, and (**F**) BCL-2 and primulasaponin.

**Figure 3 antioxidants-11-01377-f003:**
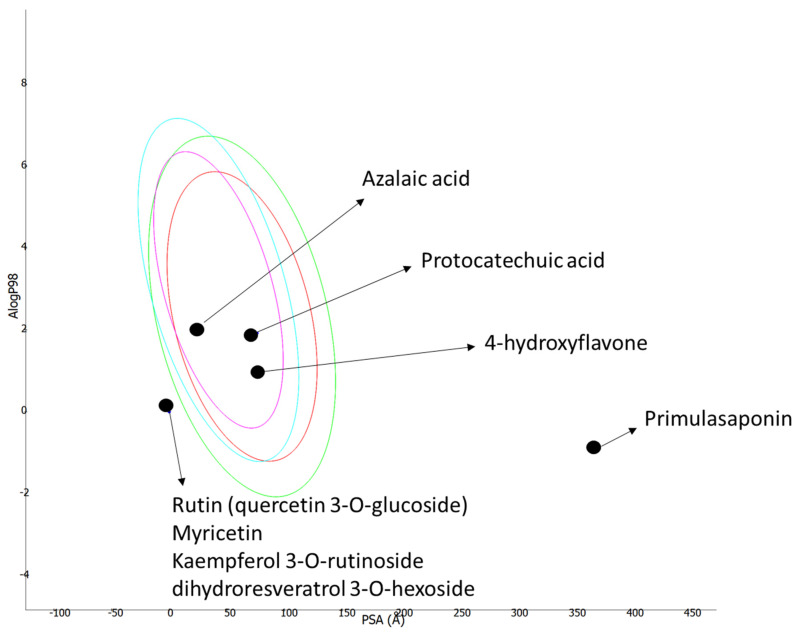
Logarithm of octanol–water partition coefficient (LogP) versus topological surface area (PSA). The four ellipses define area well-absorbed molecules are expected to fall. A total of 95 and 99% of a compound should be in ellipses colored in red and green, respectively. For the blood–brain barrier penetration, 95 and 99% of a compound should be located within the ellipses colored with magenta and aqua, respectively.

**Figure 4 antioxidants-11-01377-f004:**
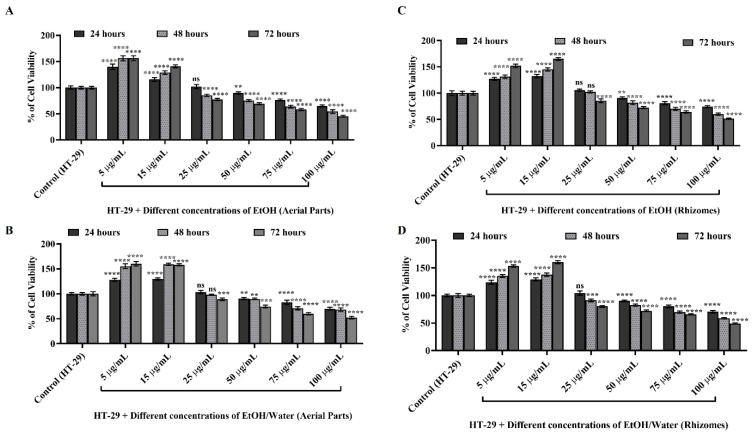
Determination of nontoxic concentration and % cell viability of EtOH and EtOH/water extracts from aerial and root parts of *Primula auriculata* in HT-29 colorectal cancer cell line with WST-1. Increasing concentrations (5–100 µg/mL) of EtOH aerial parts in (**A**), EtOH/water aerial parts in (**B**), EtOH rhizomes in (**C**), and EtOH/water rhizomes in (**D**) extracts were used to treat the HT-29 cell line for 24, 48, and 72 h, respectively. GraphPad Prism version 8 was used for statistical analysis, and the nonsignificant value was symbolized with ns. ** was used for *p* ≤ 0.01, *** was used for *p* ≤ 0.001, and *p* ≤ 0.0001 was shown with ****.

**Figure 5 antioxidants-11-01377-f005:**
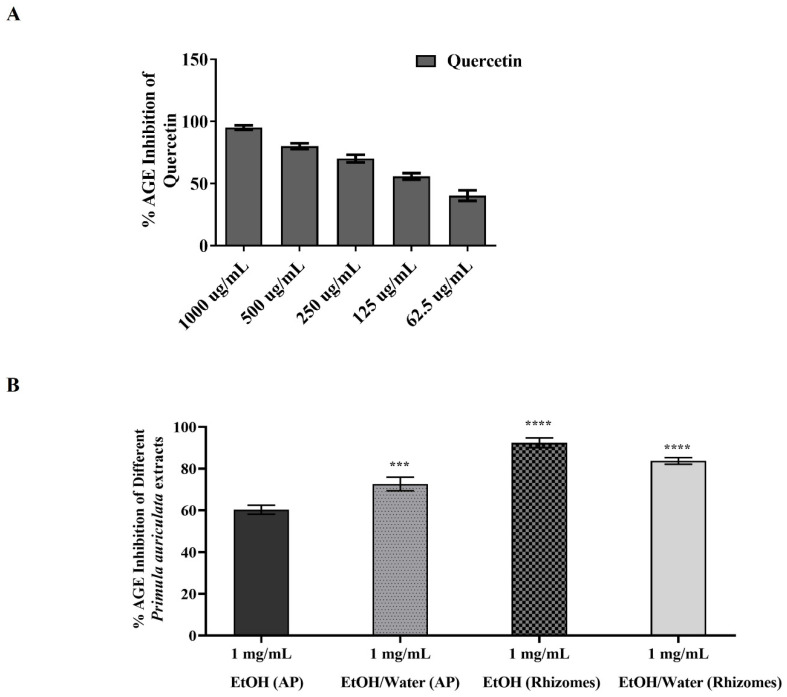
Effect of different concentrations of quercetin (in (**A**)) and 1 mg/mL of EtOH and EtOH/water extracts obtained from aerial and root parts of *Primula auriculata* (in (**B**)) on AGE formation. In (**B**), the EtOH extract obtained from aerial parts (AP) of *Primula auriculata* were compared to other extracts at the same concentration. In the statistical analysis, a *p* ≤ 0.001 value is shown with ***, and **** is indicated for *p* ≤ 0.0001.

**Figure 6 antioxidants-11-01377-f006:**
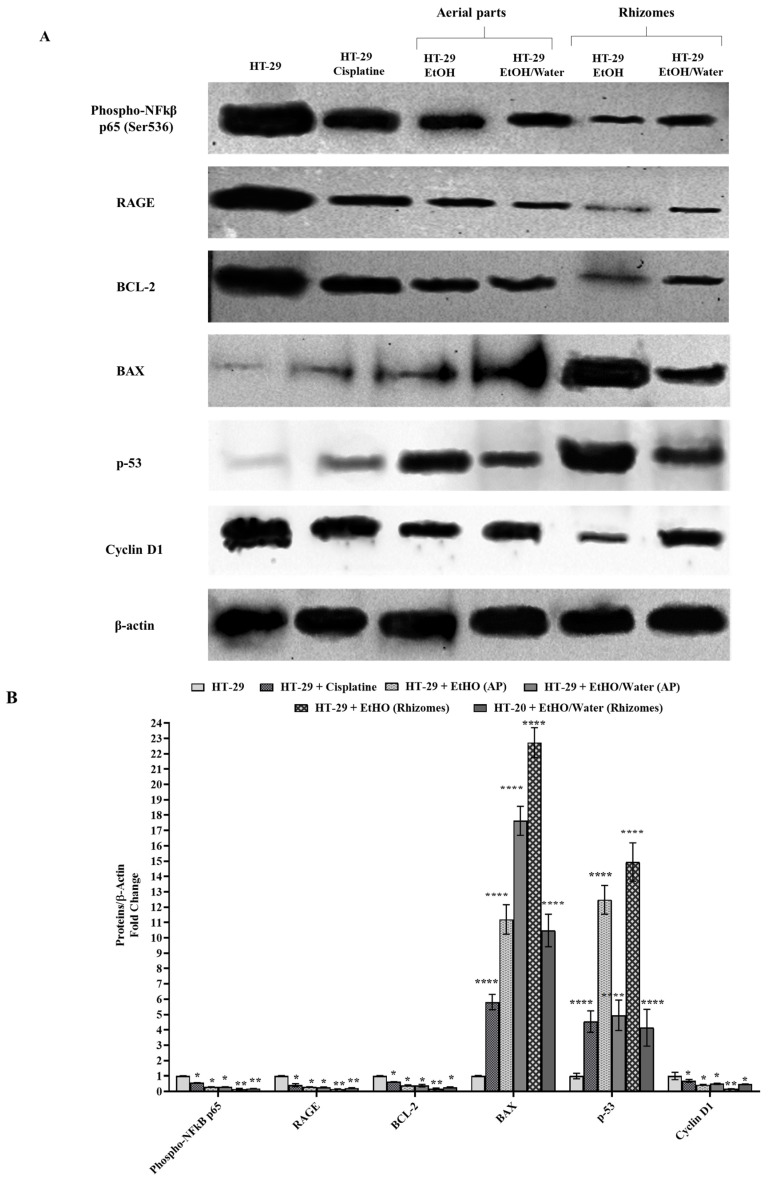
Determination of the effect of 50 µg/mL EtOH and EtOH/water extracts obtained from different parts of *Primula auriculata* on protein synthesis by Western blot. The band densities of phospho-NF-_K_β, RAGE, BCL-2, BAX, p-53, and cyclin D1 proteins on the nitrocellulose membrane were visualized by ChemiDOC in (**A**). β-actin was used as a loading control and 1-fold was assumed for comparison. In (**B**), statistical analysis was performed with GraphPad Prism 8, and * was used for *p* ≤ 0.05, ** was used for *p* ≤ 0.01, and **** was used for *p* ≤ 0.0001.

**Figure 7 antioxidants-11-01377-f007:**
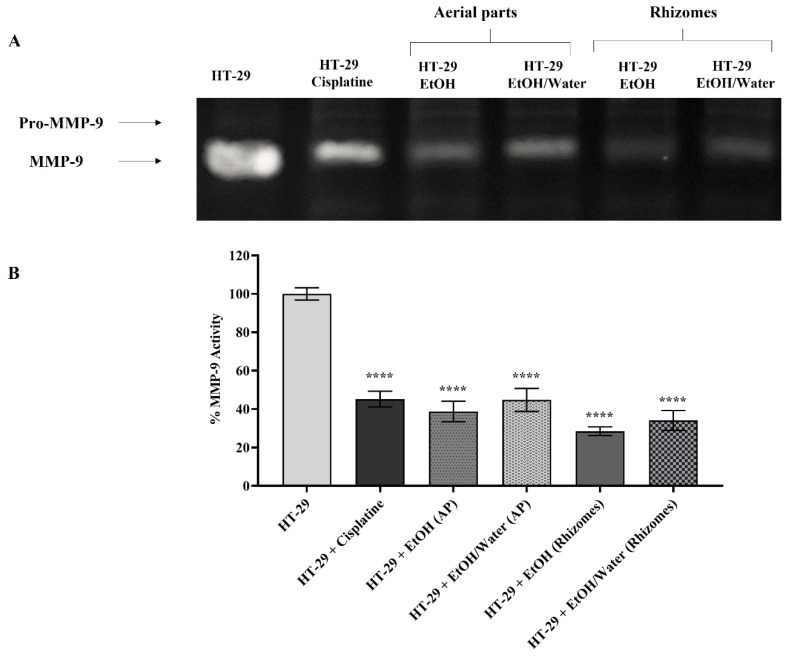
Detection of MMP-9 enzyme activity by gelatin zymography in HT-29 cell line treated with different *Primula auriculata* extracts and positive control cisplatin. Lytic white bands on the dark background of the gel stained with Coomassie brilliant blue were captured by ChemiDoc (in (**A**)). For the detection of MMP-9 enzyme activity, untreated HT-29 cells were selected as the control, and the band intensity was measured with Image J. In the statistical analysis, *p* ≤ 0.0001 is presented with **** (in (**B**)).

**Figure 8 antioxidants-11-01377-f008:**
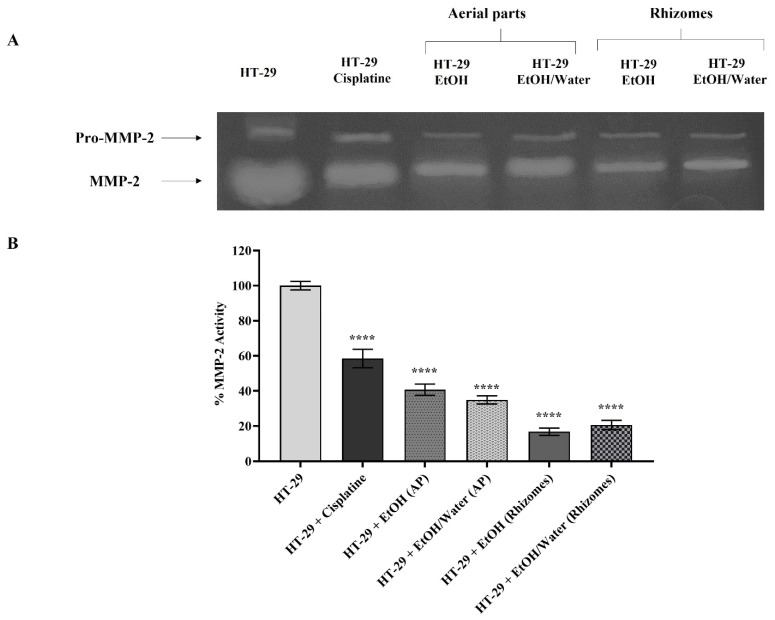
Detection of MMP-2 enzyme activity by gelatin zymography in HT-29 cell line treated with different *Primula auriculata* extracts and positive control cisplatin. Lytic white bands on the dark background of the gel stained with Coomassie brilliant blue were captured by ChemiDoc (in (**A**)). For the detection of MMP-2 enzyme activity, untreated HT-29 cells were selected as the control, and the band intensity was measured with Image J. In the statistical analysis, *p* ≤ 0.0001 is presented with **** (in (**B**)).

**Figure 9 antioxidants-11-01377-f009:**
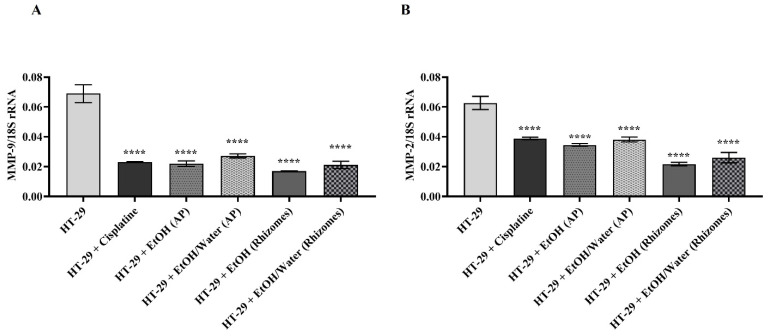
Effect of 50 µg/mL EtOH and EtOH/water extracts obtained from aerial and root parts of *Primula auriculata* on MMP-9 (in (**A**)) and MMP-2 (in (**B**)) mRNA gene expression levels. The 18sRNA gene was used as a reference gene to detect changes in target gene levels in the HT-29 cell line. In GraphPad Prism 8 statistical analysis, **** was used for *p* ≤ 0.0001.

**Table 1 antioxidants-11-01377-t001:** Yield (%),total phenolic (TPC) and flavonoid (TFC) contents, and total antioxidant capacity (PBD) of the tested extracts.

Plant Parts	Solvents	Yield (%)	TPC (mg GAE/g Dry Extract)	TFC (mg RE/g Dry Extract)	PBD (mmol TE/g Dry Extract)
Aerial parts	EA	1.73	32.36 ± 0.43 ^g^	9.64 ± 0.17 ^d^	1.84 ± 0.11 ^b^
	EtOH	4.15	45.77 ± 0.30 ^e^	22.84 ± 0.42 ^c^	1.76 ± 0.08 ^b^
	EtOH/Water	16.67	56.81 ± 0.23 ^c^	63.92 ± 0.41 ^a^	1.49 ± 0.16 ^c,d^
	Infusion	12.97	52.88 ± 0.66 ^d^	57.71 ± 0.56 ^b^	1.24 ± 0.01 ^d^
Rhizomes	EA	1.61	36.19 ± 0.26 ^f^	3.72 ± 0.10 ^f^	1.45 ± 0.11 ^c,d^
	EtOH	7.64	96.66 ± 0.61 ^a^	9.64 ± 0.67 ^d^	2.59 ± 0.13 ^a^
	EtOH/Water	14.87	70.51 ± 0.34 ^b^	7.72 ± 0.03 ^e^	1.92 ± 0.02 ^b^
	Infusion	16.06	46.08 ± 0.17 ^e^	6.85 ± 0.13 ^e^	1.67 ± 0.04 ^b,c^

Values are reported as means ± S.D. of three replications. GAE: Gallic acid equivalents; RE: Rutin equivalents; TE: Trolox equivalents. EA: Ethyl acetate; EtOH: Ethanol. Different letters indicate significant differences in the tested extracts (*p* < 0.05).

**Table 2 antioxidants-11-01377-t002:** Specialized metabolites in *Primula auriculata* extracts.

No.	Identified/Tentatively Annotated Compound	Molecular Formula	Exact Mass [M-H]	Fragmentation Pattern in (-) ESI-MS/MS	t_R_ (Min)	Δ ppm	Distribution	Level of Identification (CAWG)
Phenolic acids and glycosides								
1.	protocatechuic acid	C_7_H_6_O_4_	153.0181	153.0181 (17.85), 109.0280 (100)	2.01	−7.790	1,2,3,4,5,6,7,8	1
2.	caffeic acid	C_9_H_8_O_4_	179.0341	179.0338 (22.92), 135.0437 (100), 107.0488 (2.38)	3.54	−4.926	1,2,3,4,7,8	1
3.	neochlorogenic (3-caffeoylquinic) acid	C_16_H_18_O_9_	353.0873	353.0873 (39.89), 191.0551 (100), 179.0338 (65.22), 135.0438 (67.15)	2.37	−1.488	1,2,3,4,5,6,7,8	1
4.	chlorogenic (5-caffeoylquinic) acid	C_16_H_18_O_9_	353.0880	353.0880 (4.84), 191.0552 (100), 179.0334 (0.93), 161.0233 (2.30), 135.0436 (0.85)	3.19	0.665	1,2,3,4,5,6,7,8	1
5.	primeverin/primulaverin	C_20_H_28_O_13_	475.1468	521.1525 (11.02), 475.1468 (4.90), 443.1199 (12.31), 293.0880 (100), 233.0674 (1.33), 181.0500 (10.28), 166.0261 (30.25), 149.0444 (9.37), 131.0332 (7.67)	4.70	2.328	2,3,5,6,7,8	2
Flavones and flavonols								
6.	4′-hydroxyflavone	C_15_H_10_O_3_	237.0555	237.0552 (100), 209.0603 (7.98), 117.0331 (65.16)	9.42	−1.044	1,2,3,4,5,6,7,8	1
7.	7,4′-dihydroxyflavone	C_15_H_10_O_4_	253.0504	253.0504 (100), 225.0538 (1.05), 209.0609 (0.99), 133.0281 (51.31), 117.0330 (1.00)	7.94	−0.799	1,2,3,4,5,6,7	2
8.	7,3′,4′-trihydroxyflavone	C_15_H_10_O_5_	269.0458	269.0458 (100), 241.8075 (0.35), 225.0551 (2.66), 201.0185 (13.80), 173.0233 (3.58), 153.0177 (0.50), 135.0073 (37.58) (1,3A), 133.0281 (25.40) (1,3B), 91.0173 (0,4A)	8.17	0.867	1,2,3	2
9.	apigenin	C_15_H_10_O_5_	269.0454	269.0454 (100), 225.0555 (2.26), 204.9892 (0.99), 151.0022 (6.20), 149.0231 (5.05), 117.0330 (19.33), 107.0123 (4.96)	8.60	−0.396	1,2,3	1
10.	acacetin	C_16_H_12_O_5_	283.0612	283.0612 (100), 268.0378 (67.81), 211.0401 (1.79), 151.0023 (2.29), 133.0284 (0.48), 107.0122 (0.48)	11.35	0.718	1,2,3,4	2
11.	luteolin	C_15_H_10_O_6_	285.0405	285.0404 (100), 257.0452 (0.37), 241.0504 (0.25), 151.0024 (4.61), 133.0281 (23.07), 107.0123 (4.33)	7.56	0.136	1,2,3,4,5,6,7	1
12.	kaempferol	C_15_H_10_O_6_	285.0405	285.0404 (100), 178.9914 (0.49), 161.7458 (0.48), 151.0025 (2.07), 117.0331 (0.92), 107.0122 (0.98)	8.82	0.031	1,2,3,4,5,6,7,8	1
13.	quercetin	C_15_H_10_O_7_	301.0355	301.0353 (100), 273.0403 (2.68), 257.0466 (1.25), 178.9975 (19.30), 161.0230 (0.40), 151.0024 (44.12), 121.0280 (13.69), 107.0123 (14.49)	7.60	0.279	1,2,4,5,6,7,8	1
14.	cirsimaritin	C_17_H_14_O_6_	313.0719	313.0719 (100), 298.0484 (60.56), 283.0252 (2.28), 255.0291 (1.74), 163.0027 (1.02), 149.9946 (14.61), 133.0282 (20.31), 1798.9915 (1.35), 151.0023 (0.69)	11.00	0.539	1,3	2
15.	dihydroxy-dimethoxy flavone	C_17_H_14_O_6_	313.0714	313.0714 (11.33), 298.0483 (100), 283.0245 (17.27), 255.0301 (3.48), 133.0280 (8.28), 116.9938 (1.29)	11.18	−1.122	1,3,4	3
16.	rhamnetin	C_16_H_12_O_7_	315.0507	315.0507 (90.55), 301.0298 (7.33), 300.0273 (100), 272.0330 (4.37), 255.0293 (2.53), 227.0333 (2.38), 165.9898 (12.11)	8.80	−0.971	1,2,3	1
17.	isorhamnetin	C_16_H_12_O_7_	315.0514	315.0511 (100), 301.0323 (3.08), 300.0275 (47.91), 151.0021 (11.36), 107.0121 (9.85)	9.10	1.156	1,3	1
18.	2′-hydroxyflavone-7-*O*-hexoside	C_21_H_20_O_9_	415.1037	415.1036 (13.39), 253.0503 (100), 225.0549 (1.28), 209.0602 (1.08), 133.0280 (32.83), 117.0331 (1.34)	6.23	0.662	1	3
19.	kaempferol 3-*O*-pentoside	C_20_H_18_O_10_	417.0830	417.0830 (100), 285.0397 (10.33), 284.0326 (41.06), 255.0297 (31.74), 227.0339 (26.28), 211.0378 (2.32)	6.09	0.624	1,2,3,4	2
20.	kaempferol 7-*O*-pentoside	C_20_H_18_O_10_	417.0830	417.0830 (100), 285.0405 (63.95), 284.0326 (66.41), 255.0298 (44.05), 227.0348 (31.22)	6.33	0.696	1	2
21.	kaempferol 7-*O*-deoxyhexoside	C_21_H_20_O_10_	431.0984	431.0984 (100), 285.0403 (70.22), 284.0327 (54.75), 255.0298 (6.51), 227.0345 (31.17)	6.58	0.186	1,2,3,4	2
22.	apigenin 7-*O*-glucoside	C_21_H_20_O_10_	431.0986	431.0982 (16.52), 269.0456 (100), 227.0336 (2.06), 241.0504 (0.89), 150.9850 (0.50), 133.0280 (11.87), 117.5589 (0.59)	7.69	0.464	1,2,3,4	1
23.	quercetin 3-*O*-pentoside	C_20_H_18_O_11_	433.0774	433.0774 (100), 301.0350 (84.54), 300.0275 (86.50), 271.0251 (39.29), 255.0300 (19.52), 227.0324 (2.46), 178.9983 (1.91), 151.0022 (7.32), 107.0123 (2.35)	5.74	−0.565	1,2	2
24.	luteolin 7-*O*-glucoside	C_21_H_20_O_11_	447.0932	447.0932 (100), 285.0403 (90.49), 284.0326 (37.15), 227.0344 (1.65), 151.0022 (4.77), 133.0283 (3.73), 107.0122 (1.81)	5.35	−0.122	1,2,3,4	1
25.	luteolin 3′-*O*-hexoside	C_21_H_20_O_11_	447.0936	447.0927 (100), 285.0400 (22.81), 284.0326 (56.02), 255.029 (43.74) 227.0344 (40.57), 211.0397 (0.62), 151.0389 (5.68), 133.0283 (3.21), 107.0486 (0.63)	5.90	0.773	1,2,5,6,8	2
26.	kaempferol 3-*O*-glucoside	C_21_H_20_O_11_	447.0935	447.0935 (100), 285.0393 (16.96), 284.0327 (51.53), 255.0298 (41.70), 227.0345 (40.54), 211.0412 (0.87), 151.0023 (1.10)	5.63	0.571	1,2,3,4,5,6,7,8	1
27.	luteolin 3′-*O*-pentoside	C_21_H_20_O_11_	447.0936	447.0927 (100), 285.0400 (22.81), 284.0326 (56.02), 255.02977 (43.74), 227.0344 (40.57), 211.0397 (0.62), 151.0389 (5.68), 133.0283 (3.21), 107.0486 (0.63)	5.90	0.773	1,2,5,6,8	2
28.	myricetin 3-*O*-pentoside	C_20_H_18_O_12_	449.0726	449.0716 (100), 317.0298 (28.80), 316.0223 (87.23), 287.0207 (14.49), 271.0246 (24.62), 178.9969 (2.77), 151.0019 (3.07)	5.01	0.113	1,2,3,4	2
29.	7,4′-dihydroxyflavone O-acetylhexoside	C_23_H_22_O_10_	457.1147	457.1142 (24.71), 397.0938 (7.06), 295.0595 (0.88), 253.0504 (100), 225.0548 (1.59), 209.0607 (1.28), 117.0326 (1.09), 133.0281 (37.57)	7.06	1.597	1,2,3,4,5,6,7,8	3/4
30.	myricitrin (myricetin 3-*O*-rhamnoside)	C_21_H_20_O_12_	463.0886	463.0884 (100), 317.0295 (22.04), 316.0224 (84.26), 287.0198 (15.13), 271.0248 (28.78), 178.9973 (2.76), 151.0027 (5.24), 107.0123 (1.22)	5.10	3.113	1,2,3,4,5,6,7,8	1
31.	isoquercitrin (quercetin 3-*O*-glucoside)	C_21_H_20_O_12_	463.0888	463.0883 (100), 301.0349 (41.27), 300.0274 (71.68), 271.0248 (3.70), 151.0026 (6.10), 107.0121 (2.54)	5.27	1.319	1,2,3,4,5,6,7,8	1
32.	7,4′-dihydroxyflavone O-hexoside-*O*-ethylmaleate	C_27_H_28_O_13_	559.1457	559.1469 (0.85), 457.1139 (4.62), 415.1035 (47.92), 253.0504 (100), 225.0548 (1.51), 209.0604 (1.70), 133.0281 (29.38), 117.0328 (0.59)	7.07	−0.043	1,2,3,4,5,6,7,8	4
33.	acaciin (acacetin 7-*O*-rutinoside)	C_28_H_32_O_14_	591.1730	591.1726 (16.76), 284.0646 (8.43), 283.0612 (100), 269.0413 (3.92), 268.0377 (43.72), 267.0296 (0.21), 240.0430 (1.02), 239.0346 (0.65), 151.0025 (0.51), 107.0125 (0.18)	7.58	1.051	1,2,3,5	1
34.	kaempferol 4′-*O*-rutinoside	C_27_H_30_O_15_	593.1516	593.1516 (85.52), 285.0406 (100), 284.0329 (12.77)	5.21	0.601	1,2,3,4,6,8	2
35.	kaempferol 3-*O*-rutinoside	C_27_H_30_O_15_	593.1514	593.1517 (100), 285.0400 (26.30), 284.0327 (54.97), 255.0298 (34.80), 227.0346 (22.41), 211.0393 (1.14), 151.0023 (2.39), 107.0123 (1.20)	5.40	0.399	1,2,3,4,6,8	1
36.	kaempferol 7-*O*-rutinoside	C_27_H_30_O_15_	593.1515	593.1523 (100), 285.0406 (57.00), 284.0323 (34.01), 255.0307 (32.37), 227.0347 (10.38)	5.63	0.500	1,2,3,4,5,6,7,8	2
37.	rutin (quercetin 3-*O*-rutinoside)	C_27_H_30_O_16_	609.1467	609.1468 (100), 301.0349 (31.23), 300.0276 (76.39), 271.0249 (40.61), 255.0297 (17.68), 178.9976 (3.20), 151.0024 (5.81), 107.0123 (2.36)	5.00	1.120	1,2,3,4,5,6,7,8	1
38.	pectolinarin (pectolinarigenin 7-*O*-rutinoside)	C_29_H_34_O_15_	621.1831	621.1830 (11.33), 313.0720 (100), 298.0484 (19.01), 284.0285 (2.59), 283.0249 (33.40), 255.0300 (6.17), 227.0341 (1.49), 163.0025 (2.87), 117.0329 (1.36)	7.66	0.993	1,2,3,4,5,6,7	1
39.	myricetin 3-*O*-rutinoside	C_27_H_30_O_17_	625.1418	625.1418 (100), 317.0292 (14.25), 316.0224 (72.60), 287.0196 (18.02), 271.0247 (22.26), 178.9972 (1.95), 151.0024 (6.58), 107.0116 (1.20)	4.45	1.324	1,2,3,4,5,6,8	2
40.	kaempferol 3-*O*-hex-deoxyhex-deoxyhex	C_33_H_40_O_19_	739.2093	739.2108 (86.61), 285.0405 (100), 284.0326 (53.84), 255.0297 (56.20), 227.0346 (39.82), 211.0399 (3.86), 151.0027 (2.49), 135.0073 (1.92), 107.0125 (3.60)	5.31	0.254	1,2,3,4,5,6,7,8	2
41.	quercetin 3-*O*-hex-deoxyhex-hex	C_33_H_40_O_20_	755.2040	755.2049 (100), 301.0348 (46.03), 300.0275 (78.15), 271.0248 (54.98), 255.0298 (25.98), 227.0339 (4.48), 151.0028 (10.89), 107.0120 (2.11)	4.24	0.539	1,2,3,4	2
42.	kaempferol 3-*O*-hex-hex-hex	C_33_H_40_O_21_	771.1992	771.1996 (51.83), 285.0405 (100), 255.0295 (18.63), 227.0348 (12.85), 211.0397 (3.88), 151.0021 (1.07), 107.0122 (3.19)	4.71	0.297	1,2,3,4,5,6,7,8	2
43.	quercetin 3-*O*-hex-hex-hex	C_33_H_40_O_22_	787.1943	787.1937 (83.60), 301.0351 (100), 300.0278 (79.80), 271.0247 (56.82), 255.0295 (20.25), 151.0024 (18.60), 107.0123 (3.69)	4.24	0.539	1,2,3,4,5,7	2
Chalcones and dihydochalcones								
44.	hydroxychalcone	C_15_H_12_O_2_	223.0760	223.0759 (100), 205.0648 (0.38), 202.9935 (2.10), 195.0807 (49.88), 182.9862 (2.59), 145.0281 (1.59), 119.0487 (0.58), 117.0331 (16.14), 93.0330 (8.74)	11.93	−1.896	1,4	3
45.	dihydroxychalcone	C_15_H_12_O_3_	239.0709	239.0710 (28.19), 195.0808 (1.34), 121.0282 (1.08), 119.0487 (100), 117.0332 (3.36), 93.0330 (33.11)	13.19	−1.872	1,2,3,4,5,6,7,8	3
46.	dihydroxydihydrochalcone	C_15_H_14_O_3_	241.0868	241.0867 (20.44), 197.0960 (1.02), 147.0438 (7.62), 135.0437 (100), 121.0276 (0.18), 119.0486 (0.34), 117.0337 (0.34), 93.0330 (44.52)	12.87	−0.820	1,2,3,4,5,6,7	3
47.	trihydroxychalcone	C_15_H_12_O_4_	255.0663	255.0661 (25.50), 211.0757 (100), 183.0809 (0.79), 169.0648 (3.71), 143.0490 (4.45), 119.0486 (1.08), 93.0329 (1.86)	9.63	−0.008	1,3,4,7,8	3
48.	trihydroxydihydrochalcone	C_15_H_14_O_4_	257.0818	257.0818 (100), 213.0914 (94.48), 171.0802 (1.46), 151.0391 (0.27), 121.0280 (10.62), 117.5003 (0.22), 107.0487 (43.67), 93.0330 (4.79)	9.71	−0.670	2,3,4,5,6,7,8	3
49.	tetrahydroxychalcone	C_15_H_12_O_5_	271.0616	271.0609 (100), 151.0024 (65.21), 119.0490 (43.91), 107.0127 (18.87), 93.0330 (6.17)	8.57	1.414	2,3,7,8	3
50.	tetrahydroxydihydrochalcone	C_15_H_14_O_5_	273.0769	273.0769 (100), 229.0866 (55.68), 121.0280 (17.11), 137.0231 (7.82), 107.0487 (44.35), 93.0329 (1.50)	8.43	0.305	2,3,4,7	3
Saponins								
51.	primulasaponin isomer I(primula acid I)	C_54_H_88_O_23_	1103.5644	1103.5649 (100), 957.5015 (0.8), 923.5041 (2.8), 795.4615 (0.3), 553.3922 (0.5), 455.3522 (0.9), 437.3407 (0.3), 407.3314 (1.0)	10.17	0.524	1,2,3,4,5,6,7,8	2
52.	primulasaponin(primula acid I) isomer II	C_54_H_88_O_23_	1103.5644	1103.5647 (100), 923.5104 (2.8), 795.4513 (0.5), 455.3532 (1.9)	11.99	0.306	1,2,3,4,5,6,7,8	2
53.	priverosaponin B isomer I	C_54_H_88_O_24_	1119.5593	1119.5597 (100), 973.5049 (0.6), 957.5066 (0.6), 939.4974 (3.1), 811.4457 (0.4), 775.4327 (0.1), 749.4576 (0.1), 569.3849 (1.0), 473.3637 (0.9), 423.3281 (0.8), 407.2965 (0.3), 391.3006 (0.5)	8.17	0.369	1,2,3,4,5,6,7,8	2
54.	priverosaponin B isomer II	C_54_H_88_O_24_	1119.5593	1119.5599 (100), 939.4963 (3.3), 811.4561 (0.3), 775.4206 (0.1), 569.3799 (0.9), 473.3624 (2.0), 423.2854 (0.3)	8.94	0.593	1,2,3,4,5,6,7,8	2
55.	priverosaponin B isomer III	C_54_H_88_O_24_	1119.5593	1119.5603 (100), 973.4953 (0.3), 939.4949 (3.0), 811.4513 (0.3), 569.3843 (1.0), 473.3639 (1.1), 423.3256 (0.4)	10.15	0.914	1,2,3,4,5,6,7,8	2
56.	priverosaponin B isomer IV	C_54_H_88_O_24_	1119.5593	1119.5601 (100), 939.4949 (3.5), 569.3826 (1.4), 423.3278 (0.7), 405.3184 (0.4)	10.99	0.700	1,2,3,4,5,6,7,8	2
57.	primacrosaponin isomer I	C_54_H_88_O_25_	1135.5542	1135.5552 (100), 955.4901 (3.9)	8.09	0.871	1,2,3,4,5,6,7,8	2
58.	primacrosaponin isomer II	C_54_H_88_O_25_	1135.5542	1135.5551 (100), 955.4885 (3.8), 585.3813 (0.5), 489.3595 (0.6)	8.42	0.756	1,2,3,4,5,6,7,8	2
59.	priverosaponin B 22 acetate isomer I	C_56_H_90_O_25_	1161.5698	1161.5706 (100), 981.5074 (2.9), 611.3977 (1.2), 515.3732 (1.2)	9.48	0.0.619	1,2,3,4,5,6,7,8	2
60.	priverosaponin B 22 acetateisomer II	C_56_H_90_O_25_	1161.5698	1161.5706 (100), 1101.5438 (0.2), 1015.5159 (0.6), 981.5071 (2.8), 853.4595 (0.5), 611.3957 (1.0), 583.4039 (0.5), 515.3780 (1.0), 405.3154 (0.2)	9.65	0.619	1,2,4,5,6,7,8	2
61.	priverosaponin B 22 acetateisomer III	C_56_H_90_O_25_	1161.5698	1161.5703 (100), 1015.5275 (0.6), 981.5071 (3.1), 853.4584 (0.5), 611.3962 (0.6), 513.3572 (1.6)	9.92	0.404	1,2,3,4,5,6,7,8	2
62.	priverosaponin B 22 acetateisomer IV	C_56_H_90_O_25_	1161.5698	1161.5701 (100), 981.5082 (2.8), 853.4612 (0.1), 835.4540 (0.1), 817.4373 (0.1), 673.4022 (0.1), 611.4022 (0.9), 583.4010 (0.5), 515.3792 (0.6), 473.3643 (1.6)	10.49	0.197	1,2,3,4,5,6,7,8	2
Other								
63.	azelaic acid	C_9_H_16_O_4_	187.0966	187.0966 (49.74), 125.0957 (100), 123.0800 (1.46), 97.0642 (6.88)	6.30	−5.250	1,2,3,4,5,6,7,8	2
64.	quinic acid	C_7_H_12_O_6_	191.0553	191.0561 (100), 173.0450 (1.72), 127.0387 (4.07), 111.0440 (1.41), 93.0331 (5.06), 85.0279 (18.16)	3.18	−5.032	1,2,3,4,5,6,7,8	2
65.	dihydroresveratrol 3-*O*-hexoside	C_20_H_24_O_8_	391.1402	391.1401 (6.24), 229.0865 (100), 137.0229 (6.22), 122.0359 (45.14), 108.0201 (17.96), 93.0329 (2.61)	4.38	0.995	1,2,3,5,6,7,8	3

1 ÷ 4 aerial parts extracts (1—ethylacetate; 2—ethanol; 3—ethanol/water; 4—infusion); 5 ÷ 8 rhizomes extracts (5—ethylacetate; 6—ethanol; 7—ethanol/water; 8—infusion).

**Table 3 antioxidants-11-01377-t003:** Radical scavenging (1,1-diphenyl-2-picrylhydrazyl (DPPH) and 2,2′-azino-bis(3 ethylbenzothiazoline) 6 sulfonic acid (ABTS)), reducing power (cupric-ion-reducing antioxidant capacity (CUPRAC) and ferric-ion-reducing antioxidant power (FRAP)), and metal-chelating ability (MCA) of the tested extracts.

Plant Parts	Solvents	DPPH (mg TE/g Dry Extract)	ABTS (mg TE/g Dry Extract)	CUPRAC (mg TE/g Dry Extract)	FRAP (mg TE/g Dry Extract)	MCA (mg EDTAE/g Dry Extract)
Aerial parts	EA	5.41 ± 0.23 ^g^	38.40 ± 0.95 ^e^	63.15 ± 0.54 ^g^	26.55 ± 0.11 ^h^	3.96 ± 0.48 ^g^
	EtOH	52.22 ± 1.58 ^f^	75.51 ± 1.61 ^e^	136.92 ± 7.41 ^e^	68.96 ± 0.21 ^f^	6.66 ± 0.24 ^f^
	EtOH/Water	132.65 ± 2.91 ^c^	180.87 ± 3.72 ^c^	172.46 ± 1.26 ^c^	108.37 ± 0.70 ^c^	11.75 ± 0.28 ^d^
	Infusion	106.89 ± 3.01 ^d^	158.77 ± 8.98 ^c^	148.70 ± 1.75 ^d^	95.40 ± 0.46 ^d^	25.21 ± 0.09 ^a^
Rhizomes	EA	47.85 ± 0.07 ^f^	79.02 ± 0.26 ^d^	99.65 ± 1.14 ^f^	51.08 ± 0.15 ^g^	13.40 ± 0.60 ^c^
	EtOH	415.06 ± 1.36 ^a^	638.30 ± 21.10 ^a^	477.77 ± 3.34 ^a^	301.02 ± 1.00 ^a^	3.90 ± 0.09 ^g^
	EtOH/Water	297.12 ± 1.43 ^b^	457.89 ± 7.97 ^b^	327.21 ± 2.55 ^b^	209.70 ± 2.13 ^b^	8.06 ± 0.58 ^e^
	Infusion	97.18 ± 1.19 ^e^	166.73 ± 5.06 ^c^	142.74 ± 3.66 ^d,e^	81.50 ± 0.25 ^e^	21.95 ± 0.15 ^b^

Values are reported as means ± S.D. of three replications. TE: Trolox equivalents; EDTAE: EDTA equivalents. EA: Ethyl acetate; EtOH: Ethanol. Different letters indicate significant differences in the tested extracts (*p* < 0.05).

**Table 4 antioxidants-11-01377-t004:** Cholinesterases (acetylcholinesterase (AChE) and butyrlcholinesterase (BChE)), tyrosinase, amylase, and glucosidase inhibitory effects of the tested extracts *.

Plant Parts	Solvents	AChE (mg GALAE/g Dry Extract)	BChE (mg GALAE/g Dry Extract)	Tyrosinase (mg KAE/g Dry Extract)	Amylase (mmol ACAE/g Dry Extract)	Glucosidase (mmol ACAE/g Dry Extract)
Aerial parts	EA	1.63 ± 0.21 ^c^	3.54 ± 0.23 ^a^	66.73 ± 0.81 ^d^	0.67 ± 0.01 ^a^	1.14 ± 0.01 ^b,c^
	EtOH	2.16 ± 0.05 ^b^	1.03 ± 0.24 ^b,c^	67.94 ± 0.19 ^c,d^	0.47 ± 0.01 ^c^	1.13 ± 0.02 ^c^
	EtOH/Water	2.22 ± 0.04 ^b^	0.48 ± 0.06 ^c,d^	73.02 ± 1.01 ^a,b^	0.44 ± 0.01 ^d^	1.16 ± 0.01 ^a^
	Infusion	1.13 ± 0.08 ^d^	na	27.92 ± 3.40 ^e^	0.11 ± 0.01 ^g^	1.12 ± 0.01 ^c^
Rhizomes	EA	2.31 ± 0.07 ^b^	3.45 ± 0.44 ^a^	70.98 ± 0.17 ^b,c^	0.54 ± 0.02 ^b^	1.15 ± 0.01 ^a,b^
	EtOH	2.63 ± 0.01 ^a^	3.51 ± 0.06 ^a^	76.36 ± 0.36 ^a^	0.42 ± 0.01 ^d^	1.16 ± 0.01 ^a^
	EtOH/Water	2.61 ± 0.01 ^a^	3.39 ± 0.08 ^a^	73.64 ± 0.20 ^a,b^	0.37 ± 0.01 ^e^	1.16 ± 0.01 ^a^
	Infusion	1.17 ± 0.04 ^d^	1.49 ± 0.14 ^b^	16.58 ± 0.90 ^f^	0.18 ± 0.01 ^f^	1.15 ± 0.01 ^a,b^

* Values are reported as means ± S.D. of three replications. GALAE: Galantamine equivalent; KAE: Kojic acid equivalent; ACAE: Acarbose equivalent; na: not active. EA: Ethyl acetate; EtOH: Ethanol. Different letters indicate significant differences in the tested extracts (*p* < 0.05).

**Table 5 antioxidants-11-01377-t005:** Calculated binding energy values of extracted compounds from *P. auriculata* against protein targets, including cancer proteins (MMP-9 and BCL-2).

Compound	AChE	BChE	Tyrosinase	Amylase	Glucosidase	MMP-9	BCL-2
Kcal/mol							
protocatechuic acid	−7.14	−6.41	−6.44	−6.35	−8.12	−8.28	−4.64
kaempferol 3-*O*-rutinoside	−11.21	−11.92	−6.17	−9.95	−10.65	−11.35	−10.04
rutin (quercetin 3-*O*-glucoside)	−13.89	−11.63	−6.23	−9.88	−10.33	−9.62	−8.8
4′-hydroxyflavone	−4.58	−4.38	−5.41	−5.33	−4.87	−3.57	−5.12
4-hydroxychalcone	−8.87	−6.89	−5.78	−6.88	−6.90	−6.09	−6.22
myricitrin (myricetin 3-*O*-rhamnoside)	−11.35	−7.70	−6.21	−6.71	−12.01	−10.00	−5.81
primulasaponin	-	-	-	-	-	−8.79	−10.41
azelaic acid	−6.25	−4.66	−6.56	−4.78	−6.82	−9.00	−2.17
dihydroresveratrol 3-*O*-hexoside	−12.85	−8.19	−5.70	−6.87	−9.67	−8.02	−5.00

**Table 6 antioxidants-11-01377-t006:** Drug-like and ADMET properties of the most abundant compounds in *P. auriculata* extracts predicted using the SwissADME server.

Compound	GI Absorption	BBB Permeant	CYP1A2	CYP2C19	CYP2C9	CYP2D6	CYP3A4	Lipinksi Rule	PAINS
protocatechuic acid	High	No	No	No	No	No	No	Yes; 0 violation	0 alerts
kaempferol 3-*O*-rutinoside	Low	No	No	No	No	No	No	No; 3 violations: MW > 500; NorO > 10; NHorOH > 5	0 alerts
rutin (quercetin 3-*O*-glucoside)	Low	No	No	No	No	No	No	No; 3 violations: MW > 500; NorO > 10; NHorOH > 5	1 alert: catechol amine
4′-hydroxyflavone	High	Yes	No	No	No	No	No	Yes; 0 violation	0 alerts
hydroxychalcone	High	Yes	No	No	No	No	No	Yes; 0 violation	1 alert: Michael acceptor
myricitrin (myricetin 3-*O*-rhamnoside)	Low	No	No	No	No	No	No	No; 2 violations: NorO > 10; NHorOH > 5	1 alert: catechol amine
Primulasaponin	Low	No	No	No	No	No	No	No; 3 violations: MW > 500; NorO > 10; NHorOH > 5	
azelaic acid	High	Yes	No	No	No	No	No	Yes; 0 violation	0 alerts
dihydroresveratrol 3-*O*-hexoside	Low	No	No	No	No	No	No	No; 1 violation: NorO > 5	0 alerts

GI: gastrointestinal absorption; BBB: blood–brain barrier; CYP: cytochrome P450; MW: molecular weight; NorO: Nitrogen or Oxygen; NHorOH: NH- or OH groups.

## Data Availability

All of the data is contained within the article and the Appendix A.

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
