# Peer review of "An In-Depth Study on the Metabolite Profile and Biological Properties of Primula auriculata Extracts: A Fascinating Sparkle on the Way from Nature to Functional Applications"

_antioxidants, 2022, doi:10.3390/antiox11071377_

Round 1

Reviewer 1 Report

Authors present the biological activities of different solvent extracts of aerial part and rhizomes of Primula auriculata The antioxidant, enzyme inhibition and cytotoxic and gene expression analysis (phospho-Nf-κB, BCL-2, BAX, p-53, cyclin-D1 and RAGE) is presented. Manuscript is interesting, scientific sound and contains info that can be of interest to readers of journal. Even though authors have done a good work and their aim is clear, some methods and result presentation-explanation is not clearly performed. Below are some queries that authors need to respond in order for their ms  to be reconsidered for publication.

1. At Introduction section which is rather short, authors need to mention if plant can be cultivated because only then industry can utilize extracts and local farmers and economy can be developed, and in which other places besides Turkey can be found. Did authors used native wild plant or cultivated?

2. Concerning extraction, maceration is an obsolete technique for research, i would expect "green" extractions techniques to be used, or even green solvents which for sure results would be better and this could add novelty to the work. Extraction is the most important part of such works to obtain bioactive compounds, if this fails this jeopardize the whole experiment. It is well known that a mixture of ethanol water  concerning antioxidants give better results, why authors checked all the other solvents, for water i understand, what about the others. Why 10g in 200ml solvent, boiled water, please add reference/s.

3.L. 105-106 authors say that results for assays are expressed in mg GAEs/g extract and mg REs/g extract. what is the meaning of g extract? Is it per g or per extract? Authors extracted 10g right? At table 1 results are expressed per g. Starting from lyophilised extracts which solvent was used for TPC and TFC? Method and results expression are vague. For assays two decimal digits for result expression is not needed.

4. Authors say UHPLC–HRMS, however i suggest to say UHPLC-UHMR (Ultra-High Mass Range) Hybrid Quadrupole-Orbitrap™ MS if this is the instrument used. Please add reference for method used. Flow rate of solvent/s is not reported.

5. There is no connection, link between materials and methods section and Tables or figures. Also for methods used it is not clear after lyophilization how authors and in which solvent they used different plant extracts.e.g. L. 202-204 authors used different concentrations of extracts, dissolved in which solvent (remember that initially extracts were lyophilized). For section 2.9. it is clear that EtOH and EtOH/water extracts were used, but still which solvent was used to dilute lyophilized sample not mentioned..

6.  For RT-PCR authors say that: after treatment of non-toxic concentrations of EtOH and EtOH/Water extracts, not clear what is meant here, why only these solvent extracts, what about the other two?

7. conclusion section is too generic. which specific compounds, from which solvent and in which concentration are responsible for the biological activity mentioned. Concluding that extracts showed antioxidant activity and that are GE/RAGE and MMP inhibitors is too generic.

references are up to date, minor spelling and grammar mistakes.

Author Response

Reviewer 1: Corrections were performed in yellow color.

Authors present the biological activities of different solvent extracts of aerial part and rhizomes of Primula auriculata.  The antioxidant, enzyme inhibition and cytotoxic and gene expression analysis (phospho-Nf-κB, BCL-2, BAX, p-53, cyclin-D1 and RAGE) is presented. Manuscript is interesting, scientific sound and contains info that can be of interest to readers of journal. Even though authors have done a good work and their aim is clear, some methods and result presentation-explanation is not clearly performed. Below are some queries that authors need to respond in order for their ms  to be reconsidered for publication.

  1. At Introduction section which is rather short, authors need to mention if plant can be cultivated because only then industry can utilize extracts and local farmers and economy can be developed, and in which other places besides Turkey can be found. Did authors used native wild plant or cultivated?

Response: We used wild native plants in the present study. In this context, we have improved the introduction to the importance of chemical and biological studies on the wild plant species. Some recent studies have been cited in the revised version.

  1. Concerning extraction, maceration is an obsolete technique for research, i would expect "green" extractions techniques to be used, or even green solvents which for sure results would be better and this could add novelty to the work. Extraction is the most important part of such works to obtain bioactive compounds, if this fails this jeopardize the whole experiment. It is well known that a mixture of ethanol water  concerning antioxidants give better results, why authors checked all the other solvents, for water i understand, what about the others. Why 10g in 200ml solvent, boiled water, please add reference/s.

Response: Thanks for your comments. Yes, I agree with you and maceration is a traditional method. However, the technique is used to extract thermolabile components from plant materials and the technique is not specific equipment. Thus, we performed maceration to apply it in different research studies without specific equipment. In addition, previous studies on members of the genus Primula favored maceration as an extraction method. For this reason, maceration was also chosen for comparison in the present study. I agree with you and green extraction techniques such as ultrasound-assisted or microwave-assisted extraction techniques are gaining increasing interest in recent studies. However, we don’t have specific equipment for green extraction, so we proceed with maceration to complete the study. Regarding solvent selection, to our knowledge, we have little information on the chemical profiles and biological properties of t members of the genus Therefore, we designed the study using different solvents that have different polarities. At this point, we want to select the best solvent (s) for further applications. To prepare water extracts, we used the traditional infusion technique and the plant materials were kept in boiled water. The solid-solvent ratio was chosen as 1/20 and the ratio was used in our previous papers. The papers have been cited in the revised version.

3.L. 105-106 authors say that results for assays are expressed in mg GAEs/g extract and mg REs/g extract. what is the meaning of g extract? Is it per g or per extract? Authors extracted 10g right? At table 1 results are expressed per g. Starting from lyophilised extracts which solvent was used for TPC and TFC? Method and results expression are vague. For assays two decimal digits for result expression is not needed.

Response: We have expressed the results as mg standard equivalent per gram of dry extract. To avoid confusion, we have revised the experimental and results-discussion section in the revised version. Tables have also been revised for this purpose. For water extracts, the extracts were again dissolved with water after the freeze-drying process. To explain the results, we have preferred two decimal places in our previous articles, and this method is also one of the most common. You could see it in the recent papers from the journal (Pandur et al., 2022, Antioxidants,11, 1330; Andriopoulos et al., 2022, Antioxidants, 11, 1320; Cáceres-Vélez et al., 2022; Antioxidant, 11, 1280)

  1. Authors say UHPLC–HRMS, however i suggest to say UHPLC-UHMR (Ultra-High Mass Range) Hybrid Quadrupole-Orbitrap™ MS if this is the instrument used. Please add reference for method used. Flow rate of solvent/s is not reported.

Response: Indeed, Q-Orbitrap mass spectrometer belongs to the high resolution, accurate mass (HRAM) detectors (https://www.thermofisher.com). Basically, Q-Exactive Orbitrap together with LTQ Orbitrap, Q-TOF and Fourier Transorm ion cyclotron resonance (FT-ICR) forms a family of high resolution mass spectrometry analyzers (Aydogan, 2019; Wolfender et al., 2019). A quick survey of the literature in the metabolite profiling and metabolomics domain showed that the ultra high-performance liquid chromatography coupled to Q-Exactive Orbitrap is commonly named UHPLC-HRMS (Allard et al., 2016, 2017; Aydogan, 2019; Wolfender et al., 2019). On the other hand, among the main characteristics related to the performance of a mass analyzer is MS range along with efficiency, MS resolving power, MS accuracy. Beside the large-scale identification in complex samples, the same term was used in the MS-based structural elucidation of individual unknown compounds including small molecules identification (<2000 Da) (de Vijler et al., 2017). Taking into consideration the aforementioned studies and our practical routine in the field of specialized metabolite profilings, we state that the use of the term UHPLC-HRMS is appropriate and commonly adopted.

References

Allard P-M, Genta-Jouve G, Wolfender J-L. Deep metabolome annotation in natural products research: towards a various cycle in metabolite identification. Curr opin chem biol. 2017; 36:40-49.

Allard P-M, Peresse T, Bisson J, et al. (2016). Integration of molecular networking and in‐Silico MS/MS fragmentation for natural products dereplication. Anal Chem 88: 3317–3323.

Aydoğan C. Recent advances and applications in LC-HRMS for food and plant natural products: a critical review. Anal Bioanal Chem. 2020; 412:1973–1991.

Wolfender J-L, Allard P-M, Kubo M, Queiroz FE. Меtabolomics strategies for the dereplicationof polyphenols and other metabolites. . In Recent advances in polyphenol research. Eds Halbwirth H, Stich K, Cheynier V, Quideau S. Vol. 6, 1st ed, John Wiley & Sons Ltd. 2019; 183-205

De Vijlder T, Valkenborg D, Lemiere F et al. A tutorial in small molecule identification via electrospray ionization- mass spectrometry: The practical art of structural elucidation. Mass Spec Rev 2018; 37:607-629.

Section 2.4 was rephrased. The reference for the method used is provided. The data about the flow rate are embedded into the text.

  1. There is no connection, link between materials and methods section and Tables or figures. Also for methods used it is not clear after lyophilization how authors and in which solvent they used different plant extracts.e.g. L. 202-204 authors used different concentrations of extracts, dissolved in which solvent (remember that initially extracts were lyophilized). For section 2.9. it is clear that EtOH and EtOH/water extracts were used, but still which solvent was used to dilute lyophilized sample not mentioned.

Response: In all cell culture experiments presented in L. 225-226, all EtOH and EtOH/Water extracts were dissolved in serum-free DMEM (SFM DMEM), which is the growth medium for the cells. In this way, any toxicity or background effect is avoided.

           Distilled water was used to prepare the working concentration in the AGE assay protocol presented in section 2.9, and this information has been included in the revised manuscript.

           In addition, the non-toxic concentration (50 µg/mL) obtained from the WST-1 test and applied in the tests was included in the experimental setup given in L-732, L-848 and L-856.

  1. For RT-PCR authors say that: after treatment of non-toxic concentrations of EtOH and EtOH/Water extracts, (a) not clear what is meant here, (b) why only these solvent extracts, what about the other two?

Response: (a) We have rewritten the mentioned sentence to remove the ambiguity.

            “…… after treating HT-29 cells with non-toxic concentrations of EtOH and EtOH/Water extracts (50 µg/mL) obtained from the aerial parts and rhizomes of P. auriculata”.

            (b) The reason why only certain extracts were used for molecular approach in apoptotic pathway has been described in section 3.6. in detail.

           “ 3.6. Molecular approach in apoptotic pathway

As shown in Tables 2, -3 and 4, it has been observed that different extracts obtained from the aerial parts and rhizomes of P. auriculata, especially EtOH and EtOH/Water samples have rich chemical content, high antioxidant potentials and enzyme activity. As a result of these experiments, EtOH and EtOH/Water extracts obtained from the aerial parts and rhizomes of P. auriculata were used in in vitro cell culture experiments to be applied to the HT-29 colon cancer cell line. Although there are preliminary data on the potential of P. auriculata against those cell lines, the current study provides detailed information in understanding the molecular mechanisms in cancer progress such as proteins, enzymes and genes participating in apoptotic pathways. Also, the AGE/RAGE interaction after the treatment with extracts was also indicated.”

  1. conclusion section is too generic. which specific compounds, from which solvent and in which concentration are responsible for the biological activity mentioned. Concluding that extracts showed antioxidant activity and that are GE/RAGE and MMP inhibitors is too generic.

references are up to date, minor spelling and grammar mistakes.

Response: We have been revised conclusion. In addition, we have used recent literatures to improve the paper in the revision stage.

Reviewer 2 Report

In the present manuscript the authors describe for the fist time a comprehensive the metabolite profile and the biological properties of the extracts of the Primula auriculata. The manuscript flows smooth starting from a solid and detailed introduction that justifies the experimental part. The technical description and the amount and quality of the reported data is a clear point of merit. The critical evaluation and analysis to the experimental data is provided in a high-level standard. The level of accuracy in the description of the MS profiles and the investigation of the apoptotic pathway are impressive. Finally, the quality of the figures and tables, along with the large number of suitable references higher the strength of the present work. Therefore, I do recommend publication of the manuscript in the present form.

Minor point to be corrected:

Page 1 line 40 “65 compound” should be revised as “65 compounds”

Author Response

Reviewer 2: Corrections were performed in green color.

In the present manuscript the authors describe for the fist time a comprehensive the metabolite profile and the biological properties of the extracts of the Primula auriculata. The manuscript flows smooth starting from a solid and detailed introduction that justifies the experimental part. The technical description and the amount and quality of the reported data is a clear point of merit. The critical evaluation and analysis to the experimental data is provided in a high-level standard. The level of accuracy in the description of the MS profiles and the investigation of the apoptotic pathway are impressive. Finally, the quality of the figures and tables, along with the large number of suitable references higher the strength of the present work. Therefore, I do recommend publication of the manuscript in the present form.

Minor point to be corrected:

Page 1 line 40 “65 compound” should be revised as “65 compounds”

Response: Thanks for your positive comments. The correction has been performed.

Reviewer 3 Report

The aim of the work is to examine in detail the chemical composition, antioxidant and biological properties of Primula auriculata extracts. The aim of the work is not very interesting and innovative, while the work has a significant scientific value. The number of parameters studied in the work significantly increases the importance and value of the research described in the manuscript. The analytical methods are standard, the use of an appropriate detector in the chromatographic method made it possible to fully identify the compounds contained in the extract, which is also an important aspect. The drawback is that the studies were carried out only on plants from one season and one area, which gives no information about the diversity in the composition of Primula auriculata.

For the readability of the tables, I recommend that the titles also include abbreviations. The title in table 1 should look like this: Total phenolic (TPC), flavonoid (TFC) contents and total antioxidant capacity (PBD) of the tested extracts.

The description of Figure 4 also needs to be completed or explained. If “the non-significant value was symbolized with ns. ** was used for p≤ 0.01 and **** was used for p ≤ 0.0001 ****”. Does that mean 3 stars? This should be explained better and more clearly. For better readability of Figure 4, I also propose to reduce the markings, if it is possible. The markings should also be the same for all drawings. The same markings should be introduced for figures 4, 5 and 6.

The work test requires some editorial corrections (e.g. L98 - comma by dot)

The described extraction method requires some clarifications. Tissue maceration was performed overnight at room temperature. Was it carried out in a stirring or shaking device? How was the extract separated from the material before the solvents were evaporated? Was the filtration applied as in the case of water extraction or the samples were centrifuged? If filtration is used, on which filters? How the samples were evaporated. Was lyophilization used for all extracts? If so, under what conditions and in what device? Small amounts of crushed material were taken for extraction. However, it is not known what amount of material the powder was obtained from, which was then used for extraction. Completing all this information will allow the experiment to be repeated and will give an answer whether it can be considered that the research material was representative.

As the work uses many standard analytical methods, I believe that the lack of or a limited description of them is appropriate.

Some of the obtained results also require some explanation. Why for TPC in aerial part the total polyphenol content is lower than the flavonoid content?. The differences are, of course, in the analytical method, the different substances to which they are equivalent. But are these all factors? And why is such a relationship observed only in the case of Aerial parts in the two extraction methods? This should be explained.

In chapter (3.2) on the results in antioxidant activity there is insufficient discussion. It was only concluded that the extract containing the most polyphenols was characterized by the highest activity. So why so many different determinations of antioxidant activity were made if the results were not compared? Do different analytical methods produce different results? And what could it be caused by? Did also ethanol solvents cause the extracts to have the highest polyphenol content and thus the highest activity in earlier publications?

Author Response

Reviewer 3: Corrections were performed in turquoise color.

The aim of the work is to examine in detail the chemical composition, antioxidant and biological properties of Primula auriculata extracts. The aim of the work is not very interesting and innovative, while the work has a significant scientific value. The number of parameters studied in the work significantly increases the importance and value of the research described in the manuscript. The analytical methods are standard, the use of an appropriate detector in the chromatographic method made it possible to fully identify the compounds contained in the extract, which is also an important aspect. The drawback is that the studies were carried out only on plants from one season and one area, which gives no information about the diversity in the composition of Primula auriculata.

For the readability of the tables, I recommend that the titles also include abbreviations. The title in table 1 should look like this: Total phenolic (TPC), flavonoid (TFC) contents and total antioxidant capacity (PBD) of the tested extracts.

Response: We have checked and corrected the titles of tables.

The description of Figure 4 also needs to be completed or explained. If “the non-significant value was symbolized with ns. ** was used for p≤ 0.01 and **** was used for p ≤ 0.0001 ****”. Does that mean 3 stars? This should be explained better and more clearly. For better readability of Figure 4, I also propose to reduce the markings, if it is possible. The markings should also be the same for all drawings. The same markings should be introduced for figures 4, 5 and 6.

Response:    *** was used for p ≤ 0.001. This information was included and the caption for Figure 4 was updated.

The work test requires some editorial corrections (e.g. L98 - comma by dot)

Response: The correction has been performed.

The described extraction method requires some clarifications. Tissue maceration was performed overnight at room temperature. Was it carried out in a stirring or shaking device? How was the extract separated from the material before the solvents were evaporated? Was the filtration applied as in the case of water extraction or the samples were centrifuged? If filtration is used, on which filters? How the samples were evaporated. Was lyophilization used for all extracts? If so, under what conditions and in what device? Small amounts of crushed material were taken for extraction. However, it is not known what amount of material the powder was obtained from, which was then used for extraction. Completing all this information will allow the experiment to be repeated and will give an answer whether it can be considered that the research material was representative.

Response: We have improved the preparation of plant extracts. In order to understand the efficiency of the solvents used in the extraction, the extraction yields have been inserted in Table 1.

As the work uses many standard analytical methods, I believe that the lack of or a limited description of them is appropriate.

Response: Appropriate references have been added to support the protocol used. For the sake of similarity, we have not included all details of the protocols as these have been previously published.

Some of the obtained results also require some explanation. Why for TPC in aerial part the total polyphenol content is lower than the flavonoid content?. The differences are, of course, in the analytical method, the different substances to which they are equivalent. But are these all factors? And why is such a relationship observed only in the case of Aerial parts in the two extraction methods? This should be explained.

Response: Thank you for your comments. In the hydroalcoholic (ethanol/water) and water extracts from aerial parts, the total flavonoid content appears to be higher than the total phenolic content of the extracts. However, these observations require some explanations. Both total phenolic and flavonoid content were evaluated by colorimetric methods. Recently, the colorimetric determination of the total content of phenols and flavonoids in plant extracts has raised some concerns. The main reason is that not only certain compounds (phenols, phenolic acids or flavonoids) but also other compounds (protein, etc.) can react with the reagents used. Therefore, the results obtained must be corrected by further chromatographic techniques such as LC-MS or NMR. In this context, we characterized all extracts using the UHPLC-HRMS technique. In addition, Folin-Ciocalteu reagent was used in the total phenolic content assay and the total phenolic content was evaluated the equivalents of gallic acid. Regarding the total flavonoid content, AlCl3 was used and the results were expressed as the equivalents of rutin. The higher total flavonoid values ​​in the hydroalcoholic and aqueous extracts from aerial parts could be explained by a higher number of flavonoids of the quercetin moiety since the standard compound rutin has a quercetin moiety. In addition, the higher flavonoid levels of aerial parts can be explained by the polarity properties of flavonoids. The content of polar flavonoids in aerial parts can be higher than in roots and can therefore be extracted with polar solvents. In fact, in Table 2, the number of flavonoids identified was higher in the aerial part extracts than in the root extracts.

In chapter (3.2) on the results in antioxidant activity there is insufficient discussion. It was only concluded that the extract containing the most polyphenols was characterized by the highest activity. So why so many different determinations of antioxidant activity were made if the results were not compared? Do different analytical methods produce different results? And what could it be caused by? Did also ethanol solvents cause the extracts to have the highest polyphenol content and thus the highest activity in earlier publications?

Response: We have improved the discussion part on antioxidant properties in the revised version. Until now, no universal antioxidant assay existed to determine the antioxidant capacity of a plant extract. Therefore, antioxidant capacity is evaluated using multiple assays to assess the full antioxidant picture. In general, alcoholic and hydroalcoholic extracts had higher level of phenolics and thus the extracts exhibited a greater antioxidant potential when compared with non-polar extracts. Some recent papers also indicated the fact, and the paper were cited in the revised version.

Round 2

Reviewer 1 Report

Authors replied to my comments and performed all my suggestions adequately, hence, revised paper is suitable for publication in its current form.